# Probable Maximum Precipitation Estimation in a Humid Climate

Zahra Afzali-Gorouh[1], Bahram Bakhtiari[2], Kourosh Qaderi[3]

[1] M.Sc. Graduate in Water Resources Engineering, Department of Water Engineering, Collage of Agriculture, Shahid Bahonar University of Kerman, Kerman, Iran

[2] Assistant Professor, Department of Water Engineering, Collage of Agriculture, Shahid Bahonar University of Kerman, Kerman, Iran

[3] Associate Professor, Department of Water Engineering, Collage of Agriculture, Shahid Bahonar University of Kerman, Kerman, Iran

*Correspondence to*: B. Bakhtiari (drbakhtiari@uk.ac.ir)

**Abstract.** Probable maximum precipitation (PMP) estimation is necessary to design floods for optimal design of hydraulic structures. The aim of this study was the estimation of 24-hour PMP ($PMP_{24}$) using the statistical and hydro-meteorological (physical) approaches in the humid climate of Qareh-Su basin which is located in the northern part of Iran. Firstly, for the statistical estimate of PMP, the equations of empirical curves of Hershfield method were extracted. Then the Hershfield standard and revised methods were written in JAVA programming language, as a user-friendly and multi-platform application called the PMP Calculator. Secondly, a hydro-meteorological approach which is called convergence model was considered to calculate $PMP_{24}$. The depth-area-duration (DAD) curves were extracted to estimate average depth of precipitation for each storm. The results indicated that the maximum values of $PMP_{24}$ for the Hershfield standard and revised methods were obtained 448 and 201 mm, respectively. While the PMP obtained by the physical approach was 143 mm. Comparison of $PMP_{24}$ values with the maximum 24-hour precipitation demonstrated that based on performance criteria including MAE, MSE, RMSE, MAPE, r, and $R^2$, the physical approach performed better than the statistical approach and it was the most reliable estimates for PMP. Also, the accuracy of the Hershfield revised method was better than the standard method using modified $K_m$ values, and the standard method gives uselessly large PMP for construction costs.

## 1 Introduction

Intensive rainfalls and heavy floods are among the most catastrophic natural hazards which have large social consequences for communities all over the world. In order to reduce the destructive effects of these phenomena, flood risk management is essential. One of the most important components in flood risk management is the probable maximum flood (PMF) estimation. Hydrologists use PMF to design a hydrologic infrastructure in a given basin, such as major spillway, dam storage capacity, and flood protection structures. In order to compute PMF accurately, it is necessary to calculate the probable maximum precipitation (PMP). PMP has been defined as "The greatest depth of precipitation for a certain duration meteorologically possible for a given size storm area at a specific time of year (WMO, 2009)".

The World Meteorological Organization has widely proposed the use of statistical and hydro-meteorological (physics-based) approaches for estimating PMP (WMO, 2009).

A statistical procedure is a probabilistic approach that requires statistical analysis based on the historical extreme precipitation in the meteorological stations where at least 30 years of daily data are available. This procedure is mostly used for small basins up to 1,000 square kilometres (WMO, 2009).

Several statistical approaches have been used to derive PMP estimates. Among others, the Hershfield method (Hershfield, 1961, 1965) is recommended by WMO as well known method to calculate PMP. Other statistical methods widely used are multifractal (Douglas and Barros, 2002), traditional frequency analysis methods or different statistical distributions such as the generalized extreme value (GEV) (Vevikenaanden, 2015; Deshpande et al., 2008) and Fisher-Tippett and Beta distributions (Nobilis et al., 1990). The advantage of the multifractal approach is providing a formal framework to derive the value of extreme events, called the fractal maximum precipitation (FMP), independently of empirical adjustments, a site-specific application of FMP in orographic regions. It should be noted that the length of the record, the spatial resolution of rain gauge network, and the lack of uncertainty estimates constrained this method (Douglas and Barros, 2002). PMP estimation using GEV method requires potentially suitable distribution such as extreme value type 1 (EV1) or extreme value type 2 (EV2) or etc. Many studies focus on GEV and Hershfield method in different regions (Alias et al., 2013; Boota et al., 2015) these researchers recommended that the application of the GEV method was lead to underestimates the upper tail (Fernando and Wickramasuriya, 2011).

Among these approaches, the Hershfield method is the most frequently used (Singh et al., 2018; Lan et al., 2017; WMO, 1986, 2009). Basically, this approach is a frequency analysis method; it is different from traditional frequency analysis methods in two important respects. First, frequency analysis methods are used to determine the statistics of extremes and this method involves the application of the process of enveloping. Second, it focuses on a wide region, rather than a single station or single watershed, in order to seek a storm that approximates the physical upper limit of precipitation (WMO, 2009). Generally, this method is applied for quick assessment of PMP (Rakhecha et al., 1992).

Hydro-meteorological estimation approaches can be usually divided into various methods, such as a) the storm model approach that provides the PMP estimations based on the physical parameterization of precipitation process and the maximization of its components. This method emphasizes the meteorological analysis of the conditions responsible for the development of extreme precipitation (Collier and Hardaker, 1996; Beauchamp et al., 2013), b) the generalized method which is time-consuming and expensive, but it has many advantages. In this method maximum use can be made of available data in the region. Furthermore, consistency between estimates for basins in the region is maintained. In fact, this method provides the accurate estimates for an individual basin (Rakhecha et al., 1995, WMO, 2009), c) the moisture maximization method which widely used for PMP estimation, despite its modifications and improvements, the method has been criticized as being insufficiently physical as it assumes a linear relationship between precipitation and water holding the capacity of the atmosphere (WMO, 1986; Papalexiou and Koutsoyiannis, 2006; Casas et al., 2011; Micovic et al., 2015; Rouhani and Leconte, 2016), and d) the storm transportation method, where a certain occurred storms with similar climatological or

topographical condition, are transposed to the region and maximized. This method increases the sample size of historical storms for PMP estimation (Rakhecha and Singh, 2009; Rezacova et al., 2005). Therefore, each method has its own theoretical basis as well as advantages, disadvantages and applicable conditions. However, it is not always easy to use the hydro-meteorological methods in many parts of the world. For example, determining the maximum humidity content in

some places may not be effortless (WMO, 2009). For this reason, such physical-based approaches have not been completely established and need to be widely verified.

In order to comparison of the statistical and hydro-meteorological approaches that mentioned above, each of these approaches has some advantages and disadvantages. One of the major disadvantages of the statistical approach is that merely depends on the precipitation data and is unable to estimate the PMP value accurately (Soltani et al., 2014), as well as the

individual discrepancy to determinate the frequency factor; and also point estimation of PMP values versus area assessment. While, the advantages of this approach are its simple usage, and quick estimates in emergencies or when there is the absence of adequate in situ meteorological data. The disadvantage of the hydro-meteorological approach is the computational complexity and requires the strong hydro-meteorological knowledge, especially when there is insufficient storm data over the region. This approach has two main advantages including consideration of the majority of the atmospheric factors such

as dew point temperature, air temperature, wind speed and direction, humidity, and air pressure; it provides more reliable results than the statistical approach for a large basin. Furthermore, it is possible to use satellite images and run atmospheric models such as RegCM3, WRF, and MM5 with this approach (Soltani et al., 2014).

Some studies indicated that both statistical and physical approaches provide reliable estimates of the PMP (Rezacova et al., 2005; Casas et al., 2011). In some cases, the PMP value obtained from the statistical approach is about two times higher than

the estimated value of the physical approach (Desa et al., 2001; Fattahi et al., 2010; Chavan and Srinivas, 2015). Thus, these studies had opposing conclusions regarding the importance of the occurrence of rare events in the recorded period, the storm precipitation data which affect the computation of the average and standard deviation amounts, and the length of records in deriving changes in PMP. A general summary of these researches have indicated that the statistical approaches provide larger estimates of PMP, but it is proposed for areas where hourly rainfall, dew point temperature, wind speed, and vertical

radiosonde measurements are unavailable.

There are many studies using the hydro-meteorological and statistical approaches in different parts of Iran (Ghahraman, 2008; Naseri Moghadam et al., 2009; Fattahi et al., 2010; Shirdeli, 2012). Naseri Moghadam et al., (2009) estimated one-day PMP for 23 meteorological stations in the four central provinces of Iran using the Hershfield method. Their emphasis was to correct the frequency factor of the Hershfield method for these stations. The results indicated that the highest value of

the frequency factors was 7.6. In another study for the central regions of Iran, Soltani et al., (2014) estimated the PMP using statistical and physical approaches. They observed that the PMP estimated by the statistical approach were greater than those estimated by the physical approach.

In an overall conclusion, it was found that there are no generally recommended approaches for PMP estimation (WMO, 2009). Besides of the available extreme rainfall data, the choice of method depends on the geographical and meteorological

characteristics of the area. It is important to consider the hydro-meteorological and statistical approaches for each region. Therefore, further research is needed to resolve this important issue.

Extreme rainfalls and flash floods which occurred in the spring and summer seasons are the most common hazards in north of Iran including the southern Caspian region representing provinces of Mazandaran, Gilan, and Golestan. In recent years,

Golestan province experienced deadly floods in its historical data. Due to consequences of extreme precipitations and floods in this region, it is necessary to estimate the values of PMP and PMF to reduce the risk of them. The value of PMP is needed for designing irrigation and drainage channels, sewage collection and disposal systems, and the maximum amount of water entering the reservoirs in this region.

The present study was undertaken to achieve the following purposes: (i) to estimate the 24-hour PMP ($PMP_{24}$) using the

Hershfield statistical approach at 7 weather stations located in the northeast of Iran, (ii) to prepare a user-friendly and multi-platform program in JAVA for the PMP calculation using the Hershfield's standard and revised methods, (iii) to provide the $PMP_{24}$ spatial distribution maps in ArcGIS for the studied region, (iv) to determine the regions that are more likely to experience intense storms, (v) to estimate the $PMP_{24}$ using the hydro-meteorological approach, (vi) to compare the values obtained from both the hydro-meteorological and statistical approaches with the observed maximum 24-hour precipitation in

the study stations.

## 2 Materials and Methods

### 2.1 Study area and data

Qareh-Su basin is located in Golestan province in the northern parts of Iran with a humid climate. The Qareh-Su basin, with nearly 1760 km$^2$ area is one of the most important basins in the north of Iran. This area is important from the viewpoint of

the existence of different cities and villages, population densities, industrial and agricultural centres, flood, and watershed management schemes. 8% of the surface water (equal to 100 million cubic meters) in Golestan province is derived from the Qareh-Su basin. There are two main dams including Kowsar and Shast kalateh to supply water demand of agricultural and residential land located in this area. Also, it is one of the most flood-prone areas that had suffered severe floods throughout its long history, so that in recent years, many people have died in destructive floods. Over the period 1951–2013, the annual

average precipitation in this basin is 596 mm. Fig. 1 shows the location of stations and the study area.

[Figure 1 somewhere here]

Climatological data that were applied in calculating of PMP values in the physical approach were 3-hour dew point temperature, 3-hour wind speed and direction at 10 m elevation, 3-hour and monthly air pressure, 3-hour, and 24-hour precipitation. The required data such as air temperature and rainfall were taken from available climatological, rain-gauge

station and synoptic stations in the study area, but dew point temperature data was taken from an available synoptic station in the study area which is called Gorgan station. The records of six rain-gauge stations and one synoptic station located in the Qareh-Su basin were taken into account in this study (Table 1). The analysis was carried out for the duration of 33 years

ranging from 1981 to 2013. The data was obtained from Islamic Republic of Iran's Meteorological Organization (IRIMO) and Iran Water Resources Management Company.

**TABLE 1: Characteristics of different stations in study area.**

| Station | Longitude (E) | Latitude (N) | Altitude (m) | The average annual precipitation (mm) | Type |
|---|---|---|---|---|---|
| Ziarat | 54° 30' | 36° 42' | 950 | 460 | Rain-gauge station |
| Kord Kooy | 54° 07' | 36° 45' | 140 | 606 | Rain-gauge station |
| Edareh Gorgan | 54° 25' | 36° 51' | 75 | 591 | Rain-gauge station |
| Shast Kelateh | 54° 20' | 36° 44' | 150 | 735 | Rain-gauge station |
| Ghaz Mahalleh | 54° 12' | 36° 47' | 6 | 604 | Rain-gauge station |
| Siah Ab | 54° 30' | 36° 45' | -26 | 607 | Rain-gauge station |
| Gorgan | 54° 25' | 36° 54' | 13.3 | 569 | Synoptic station |

**2.2 Statistical approach**

The statistical approach as developed by Hershfield is based on the general frequency equation (WMO, 2009; Chow, 1951). The equation of this method as follows:

$$X_{PMP} = \overline{X}_n + K_m \cdot S_n \tag{1}$$

where $X_{PMP}$ is the PMP estimate for a certain station at the particular duration, $\overline{X}_n$ and $S_n$ are the average and standard deviation of the annual extreme series for a given duration, respectively. $K_m$ is frequency factor as a function of duration and average of annual maximum rainfall (the maximum depth of 24-hour precipitation in each year). In other words, $K_m$ is the

number of standard deviations to be added to the average of the annual extreme series to obtain PMP. In this approach, $K_m$ is calculated by $K_m$ charts which were extracted based on records of rainfall from around 2700 stations in the climatological observation of the United States (WMO, 2009). The Hershfield standard method is modified by Desa et al., (2001). In the modified method, $K_m$ is calculated by the following Eq. (2):

$$K_m = \frac{X_{max} - \overline{X}_{n-m}}{S_{n-m}} \tag{2}$$

where $X_{max}$ is maximum observed rainfall data, $\overline{X}_{n-m}$ and $S_{n-m}$ are the average and the standard deviation of the annual

extreme series without the largest value, respectively. In order to calculate the $PMP_{24}$ by the Hershfield revised method, first, the parameters in Eq. (1) are estimated. Then, the $K_m$ values for all the stations are mapped against the $\overline{X}_n$ values respectively and a smooth envelope curve is drawn. The $K_{envelope}$ value is picked up from the curve for each station's $\overline{X}_n$. The value of PMP for each station is then estimated using Eq. (1) by replacing $K_m$ with $K_{envelope}$ value (Alias and Takara, 2013).

**2.3 Physical approach**

There are two common physical approaches namely the mountainous and convergence models to calculate PMP (Joos et al., 2005). The convergence model is based on the physical characteristic of storm, i.e. dew point temperature, wind speed, wind

direction, and etc. The main steps to calculate PMP, using the convergence model are the selection of severe storms, producing the depth-area-duration (DAD) curves, moisture maximization, and wind maximization. A severe and widespread storm is a weather condition that leads to producing precipitation in all stations in the basin and even around the basin. ==The most severe and widespread storms are selected based on maximum discharge and maximum 24 hours rainfall data.==

Producing isohyets maps are one of the main steps in the preparation of DAD curves. Using an analysis of the storms, DAD curves can be obtained. DAD curves are also applied to generalized relations for other areas or other basins with similar climate and topographic characteristics. The first step to develop DAD curve is collecting the precipitation data for all areas in the storm.

The storm maximization factor is calculated by the moisture maximization factor multiplied by wind maximization factor.
The moisture maximization method is one of the acceptable procedures to maximize the rainfall values associated with severe storms (Rakhecha and singh, 2009). This method assumes that the atmospheric moisture would hypothetically rise up to a high value that is regarded as the upper limit of moisture and the mentioned limit is estimated from historical records of dew point temperature. After selection of severe and widespread storms and calculation of average rainfall depth for the study area, it is necessary to calculate maximum humidity source in order to maximize selected storms. By converting mean
monthly pressure data at each station to 1000 mb pressure level, the effect of topography could be ignored. Dew point temperature and maximum 12-hour persisting condition at the stations during all storm events were computed and reduced to equivalent mean sea level (MSL, i.e. 1000 mb pressure level). The moisture maximization factor (FM) is calculated by Eq. (3).

$$FM = \frac{W_m}{W_s} \qquad (3)$$

where $W_m$ is the maximum precipitable water in the 1000 to 200 mb levels, which can be obtained on the basis of the
20 maximum 12-hour duration dew point with 50-year return period and $W_s$ is the maximum precipitable water at 1000 to 200 mb levels which can be obtained on the basis of maximum 12-hour duration dew point in a simultaneous period with storm (WMO, 2009). Wind maximization is most commonly used in orographic regions when it appears that observed storm rainfall over a mountain range might vary in proportion to the speed of the moisture-bearing wind blowing against the range. The wind maximization ratio is simply the ratio of the maximum average wind speed for some specific duration and critical
direction obtained from a long record of observations, e.g. 50 or 100-years, to the observed maximum average wind speed for the same duration and direction in the storm being maximized. The wind speed maximization factor (MW) is defined by Eq. (4).

$$MW = \frac{MW_1}{MW_2} \qquad (4)$$

where $MW_1$ and $MW_2$ are the maximum wind speed with 100-year return period and the maximum persisting 12-hour wind speed during the storm, respectively (WMO, 2009). Finally, PMP is determined by the precipitation depth (R) multiplied by
30 moisture maximization and wind maximization factors based on Eq. (5).

$$PMP = FM \times MW \times R \tag{5}$$

## 2.4 Performance criteria

The performance of the statistical and physical approaches for estimating $PMP_{24}$ was judged by comparing the observed maximum 24-hour precipitation values with the corresponding average estimated $PMP_{24}$ values. This comparison was conducted based on six error statistics, in terms of mean absolute error (MAE, Eq. (6)), mean squared error (MSE, Eq. (7)),

5    root mean squared error (RMSE, Eq. (8)), mean absolute percentage error (MAPE, Eq. (9)), correlation coefficient (r), Eq. (10)), and coefficient of determination ($R^2$, Eq. (11)).

$$MAE = \frac{\sum_{t=1}^{n}|O_t - C_t|}{n} \tag{6}$$

$$MSE = \frac{\sum_{t=1}^{n}(O_t - C_t)^2}{n} \tag{7}$$

$$RMSE = \sqrt{\frac{\sum_{t=1}^{n}(O_t - C_t)^2}{n}} \tag{8}$$

$$MAPE = \frac{\sum_{t=1}^{n}|O_t - C_t|}{n} \times 100 \tag{9}$$

$$r = \frac{n(\sum(O_t \times C_t)) - (\sum O_t)(\sum C_t)}{\sqrt{(n\sum(O_t)^2 - (\sum(O_t))^2) \cdot (n\sum(C_t)^2 - (\sum(C_t))^2)}} \tag{10}$$

$$R^2 = \left(\frac{\sum_{t=1}^{n}\left(O_t - \bar{O}\right)\left(C_t - \bar{C}\right)}{N \times \sigma_{O_t} \times \sigma_{C_t}}\right)^2 \tag{11}$$

Where $O_t$ is maximum 24-hour precipitation, $C_t$ is calculated $PMP_{24}$, and n is data number. RMSE reveals the actual division among the estimated and observed values. When RMSE value is closer to or equal to zero, performance is more accurate. Also, the smaller values of MAE, MSE, and MAPE show the more accurate performance. The correlation coefficient varies

10    from +1 to -1. Complete correlation between two variables is expressed by either +1 or -1 and complete absence of correlation is represented by 0. $R^2$ varies between 0-1 that closer amount to 1 represents the better performance.

## 3 RESULTS AND DISCUSSION

### 3.1 Statistical approach

The focus of the study is the calculation of $PMP_{24}$ using the statistical and physical approach in the north of Iran. In order to calculate PMP using the statistical approach, the equations of adjustment factors of Hershfield method were extracted, based on the coefficient of determination ($R^2$). Adjustment factors that are applied in statistical estimation of PMP values are $K_m$, adjustment of average and standard deviation for the maximum observed event and for sample size, adjustment for fixed observational time intervals and area reduction curves. These equations permit estimation to be carried out rapidly using a computer. The 24-hour duration $K_m$ was gained by Eq. (12).

$$K_m = -5 \times 10^{-8} x^3 + 8 \times 10^{-5} x^2 - 0.052x + 19.794 \tag{12}$$

Where x is 24-hours mean annual maximum rainfall (mm). Thus, a user-friendly and multi-platform JAVA application, which is called PMP Calculator, was developed. This application was supported by all operating system such as Windows, Linux, and Macintosh OS X. It seems that this is the first attempt to design an application which is calculated PMP in four durations using both Hershfield standard and revised methods. Also, in order to compare PMP in all stations, this application calculates the ratio of PMP to the maximum depth of rainfall as a criterion independent of climatic conditions.

Using the PMP Calculator application, PMP can be calculated by Hershfield standard and revised methods for durations such as 5 minutes, 1, 6 and 24-hour durations. In this study, maximum 24 hours duration rainfall values for selected stations located in the north of Iran with a record length of 33 years was adopted to estimate the appropriate $K_m$ values.

Table 2 indicated the result of $PMP_{24}$ using the statistical approach in the study area, which was calculated by PMP Calculator application. A more detail analysis of the PMP in the study area could be presented using the $PMP_{24}$ isohyetal lines using the Hershfield standard and revised methods which are shown in Fig. 2. PMP values at each point in the study area could be approximated from these maps. Also, the range of PMP values and its variation was shown clearly. From Fig. 2(a), it is clear that the highest $PMP_{24}$ values for the standard Hershfield method is at the southwest parts of basin around the Kord Kooy and Ghaz Mahalleh stations which are from 450 to 430 mm, whereas the lowest PMP values is at the south-eastern parts of basin around the Ziarat where the isohyetal lines are less than 240 mm. From Fig. 2(b), the $PMP_{24}$ values resulting from the use of the Hershfield revised method are lower in south-eastern parts and higher in the western parts of the study area. Generally, the $PMP_{24}$ values resulting from both Hershfield methods decrease from west to east (Fig. 2). The results of Fig. 2 showed that the western parts of the basin, that are closer to Caspian Sea, experience more severe storms. These maps are applicable to specify the regions that are more likely to experience intense storms and such information could be useful for water resources planning and management, flood risk assessment, and catastrophe management.

The results indicated that for the Hershfield standard method, $K_m$ was found to be varied the range of 17 and 18. The minimum and maximum values for point $PMP_{24}$ were 232.4 and 447.7 mm (Table 2). Also, in the Hershfield standard method, there are substantial variations in the PMP results with the variation range of 215.3 mm and average and the standard deviation of 369.1 and 74.2 mm. It shows the effect of record length on the results of the standard approach and

substantial variation in the results causes uncertainty. In the Hershfield revised method, in order to calculate the $k_m$ values, just the maximum values were considered. It caused a considerable decrease in the $K_m$ values comparing with the Hershfield standard method.

The Generalized Extreme Value theory was used to calculate 50-year precipitation ($P_{50}$). For this purpose, the GEV model was fitted on rainfall data. The results indicated that there is a significant correlation between the standard and revised estimates of $PMP_{24}$ and $P_{50}$. The values of the coefficient of determination ($R^2$) between the standard and revised estimates of $PMP_{24}$ and $P_{50}$ were 0.97 and 0.98, respectively. The relationships of $PMP_{24}$ and $P_{50}$ are defined by Eq. (13 and 14). Since the application of the GEV model was led to underestimates the upper tail, the use of Hershfield approach was recommended.

$$PMP_{24} = 3.85(P_{50}) - 24.1 \tag{13}$$

$$PMP_{24} = 1.91(P_{50}) - 20.3 \tag{14}$$

**TABLE 2: $PMP_{24}$ values using Hershfield standard and revised methods in the study area.**

| stations | Maximum 24-hour precipitation (mm) | Mean 24-hour precipitation (mm) | Standard deviation Maximum 24-hour precipitation (mm) | (CV) (%) | Standard method | | | Modified method | | |
|---|---|---|---|---|---|---|---|---|---|---|
| | | | | | $K_m$ | $PMP_{24}$ (mm) | $\dfrac{PMP_{24}}{(P_{24})_{max}}$ | $K_m$ | $PMP_{24}$ (mm) | $\dfrac{PMP_{24}}{(P_{24})_{max}}$ |
| Siah Ab | 150.2 | 53.6 | 25.6 | 47.7 | 17.2 | 417.6 | 2.8 | 5.2 | 212.6 | 1.4 |
| Kord Kooy | 104.7 | 59.9 | 21.8 | 36.3 | 17.0 | 447.7 | 4.3 | 2.2 | 197.1 | 1.9 |
| Ziarat | 63.5 | 36.2 | 11.9 | 32.7 | 18.0 | 232.4 | 3.7 | 2.9 | 111.4 | 1.8 |
| Ghaz Mahalleh | 132.0 | 54.4 | 23.4 | 43.0 | 17.2 | 419.1 | 3.2 | 4.2 | 200.7 | 1.5 |
| Shast Kelateh | 92.0 | 51.3 | 15.2 | 29.7 | 17.3 | 321.7 | 3.4 | 3.1 | 148.3 | 1.6 |
| Edareh Gorgan | 139.0 | 47.3 | 24.2 | 51.1 | 17.5 | 395.4 | 2.8 | 5.3 | 197.1 | 1.4 |
| Gorgan | 95.0 | 50.9 | 17.2 | 33.8 | 17.3 | 350.0 | 3.7 | 2.9 | 159.7 | 1.7 |

Therefore, the corresponding values of $K_m$ for Hershfield revised method ranged from 2.2 to 5.3 and the minimum and maximum values for point $PMP_{24}$ were 111.4 to 200.7 mm. The variation range, average, and the standard deviation of the revised method is about half of the corresponding values of the standard method. In order to compare two these methods and compare stations, the ratio of areal $PMP_{24}$ to the maximum of 24 hours precipitation ($(P_{24})_{max}$), as a criterion independent of climatic conditions was used. The maximum and minimum value of the ratio of $PMP_{24}$ to $(P_{24})_{max}$ for the standard method was obtained 2.8 and 4.3 whereas these values for the revised method was obtained 1.4 and 1.9. The ratio of $PMP_{24}$ to $(P_{24})_{max}$ in the revised method is closer to 1; therefore, the results of the revised method are more rational. Finally, based on the revised method, the maximum $K_m$ of Hershfield equation in the study area was found to be 5.3. The approximated $K_m$ is in accordance with corresponding research in the Atrak watershed (Ghahraman, 2008) and (Desa et al., 2001; Desa and

Rakhecha, 2007) Malaysia. Much research has been done on $K_m$ in the standard method but all of them lead to a high estimation of PMP. In the revised method, just the maximum values were considered and caused a severe and perceptible decrease in $K_m$ values which were more rational (Desa et al., 2001). Due to considering actual rainfall in the calculation of $K_m$, the revised method was more stable results than the standard method in the study area. After calculation of storm maximization factor using the wind and moisture maximization factors, the physical $PMP_{24}$ was estimated.

## 3.2 Physical approach

In this study based on maximum discharge and the daily rainfall data with 24-hours duration obtained from Iran water resources management company and IRIMO as a reliable source, 8 storms were selected as the most severe and widespread storms during 1981 to 2013. The date of occurrence these storms have been given in Table 3. After selection of severe and widespread storms, the isohyet maps for each storm were plotted in ArcGIS 9.3. To produce the DAD curves, the area bounded by each isohyet line was calculated in ArcGIS 9.3. Based on Fig. 3 that shows the spatial distribution of precipitation during the storm of September 2008 as one of the most severe storms, the greatest amount of precipitation occurred over the western parts of basin that is nearest to sea, whereas the smallest amounts of precipitation occurred over the eastern parts of basin. Based on this figure, in the western parts of the basin, isohyet lines are found close to each other and the magnitude of the rainfall gradient increases; thus the variation of rainfall in this part of the basin was elevated. Table 4 illustrated the moisture and wind speed maximizations at 1000 mb for selected storms in Gorgan station. DAD curve for the storm of September 2008 showed that the amount of rainfall decreased with increasing area (Fig. 4). The results of the physical approach demonstrated that the storm of October 1993 was the severest storm and whiles the storm of November 2006 was revealed as the most lenient one (Table 5). In order to estimate the moisture maximization factor, $W_m$ with 50-year return period was calculated. Also, to calculate the wind speed maximization factor, based on Eq. (4), MW1 with 100-year return period was determined.

**TABLE 3: Date of 24 hours duration severe and widespread storms in the study area.**

| No. | Date of occurrence | No. | Date of occurrence |
| --- | --- | --- | --- |
| 1 | 11/12/1995 | 5 | 01/11/2013 |
| 2 | 10/29/1993 | 6 | 09/29/2008 |
| 3 | 11/09/2006 | 7 | 09/27/1995 |
| 4 | 07/17/2012 | 8 | 10/13/1991 |

**TABLE 4: The moisture and wind maximization at 1000 mb for selected storms in Gorgan station.**

| Date of occurrence | Maximum persisting 12hr dew point in 1000 mb level (˚C) | | moisture maximization factor | Maximum persisting 12hr wind (Knot) | | Wind maximization Factor | PMP Factor |
| --- | --- | --- | --- | --- | --- | --- | --- |
| | In the storm time | 50-year return period | | In the storm time | 100 year return period | | |
| 11/12/1995 | 15 | 17.5 | 1.17 | 8 | 13.6 | 1.70 | 1.98 |
| 10/29/1993 | 14.1 | 20.6 | 1.46 | 7 | 10.7 | 1.53 | 2.24 |

| | | | | | | | |
|---|---|---|---|---|---|---|---|
| 11/09/2006 | 15 | 19 | 1.27 | 8 | 10.2 | 1.28 | 1.62 |
| 07/17/2012 | 20.9 | 25 | 1.20 | 8 | 8.8 | 1.10 | 1.32 |
| 01/11/2013 | 8.1 | 11.9 | 1.47 | 12 | 16.3 | 1.36 | 2.00 |
| 09/29/2008 | 20.9 | 23.8 | 1.14 | 7 | 10.1 | 1.45 | 1.65 |
| 09/27/1995 | 19 | 23.8 | 1.25 | 6 | 9.7 | 1.62 | 2.02 |
| 10/13/1991 | 13.8 | 21.2 | 1.54 | 7 | 10.7 | 1.53 | 2.35 |

**TABLE 5: The PMP values estimated by the Physical approach for selected storms in the study area.**

| Date of occurrence | Average rainfall (mm) | PMP Factor | PMP (mm) |
|---|---|---|---|
| 11/12/1995 | 72.1 | 1.98 | 143.0 |
| 10/29/1993 | 64.0 | 2.24 | 143.1 |
| 11/09/2006 | 24.8 | 1.62 | 40.1 |
| 07/17/2012 | 91.8 | 1.32 | 120.8 |
| 01/11/2013 | 60.9 | 2.00 | 121.7 |
| 09/29/2008 | 75.7 | 1.65 | 124.6 |
| 09/27/1995 | 57.6 | 2.02 | 116.5 |
| 10/13/1991 | 59.5 | 2.35 | 139.9 |

Then wind and moisture maximization factors were estimated and the amount of PMP was calculated using the multiplication PMP factor on average rainfall in a cumulative area. Based on table 5, maximum PMP value is related to the storm that occurred at 10/29/1993 and minimum PMP value is related to the storm that occurred at 11/09/2006.

==After the calculation PMP using both approaches, the aim is the determination of the best approach to estimate PMP. Hence,== based on performance criteria, the physical approach could perform better than the statistical approach. Furthermore, between two Hershfield statistical methods, the accuracy of the revised method was better than the standard method using modified $K_m$ values (Table 6).

**Table 6: Statistical comparison between $(P_{24})_{max}$ and average estimated $PMP_{24}$ values**

| method | MAE | MSE | RMSE | MAPE | r | $R^2$ |
|---|---|---|---|---|---|---|
| Standard | 258.2 | 69090.5 | 262.9 | 241.7 | 0.8 | 0.63 |
| Revised | 64.36 | 4311 | 65.7 | 61.2 | 0.9 | 0.86 |
| Physical | 7.1 | 50.4 | 7.1 | 4.7 | - | - |

The physical approach is suitable and more reliable than the statistical approach, for consideration the physical characteristics of air mass and application of meteorological data such as dew point that is an indicator of the incoming air into the storm lead to more accurate estimates. The calculation of PMP using the physical approach is difficult because this method needs more meteorological data and must be investigated the meteorological maps in a different level of atmosphere that needs for a long time. Also, calculating PMP using the physical approach requires close cooperation between hydrologists and meteorologists. Although the application of the physical approach is preferred, use of the Hershfield revised method is recommended for quick and accurate PMP estimates when dew point temperature data were unavailable.

**4 Conclusions**

In the theory definition, the PMP is the extreme rainfall for a given duration that is physically possible over an area. In practice, these estimates are based on the steps that hydro-meteorologists use to maximize observed large storms to achieve PMP value. Therefore, there is a probability that the operational estimates of PMP may be exceeded. It is necessary to mention that the return period is the inverse of this probability which can be computed by choosing an adequate theoretical and empirical distribution such as the generalized extreme value (GEV) theory. Besides, there are physical and statistical approaches for calculation of PMP.

There are physical and statistical approaches for calculation of PMP. In this study, statistical (the Hershfield standard and revised methods) and physical (convergence model) approaches are used to calculate the 24-hour PMP over the study area. In order to calculate PMP using the Hershfield method, an application, which is called PMP Calculator, is designed. This application calculates PMP with 5 minutes, 1, 6 and 24-hour durations for the Hershfield standard and revised methods. Also, for calculation of PMP using the physical approach, after selection of the most severe and widespread storms and drawing DAD curves, moisture and wind factors are estimated. Finally, PMP for each storm is calculated. The results indicated that the maximum point $PMP_{24}$ values were 448 and 201 mm for the Hershfield standard and revised methods, respectively. While $PMP_{24}$ value using the physical approach was 143 mm. The results of the revised method come closest to the physical PMP. Due to considering physical characteristics of the air mass, the result of the physical approach was reasonable and in compliance with real rainfall over the study area.

It should be noted that all of these approaches have uncertainty in the estimation of PMP. In the statistical approach, significant uncertainty can take place from the use of the enveloping curve of the frequency factor, and uncertainty in the sample mean and standard deviation. Therefore, Hershfield's frequency factor in standard method led to overestimate PMP (448 mm). In order to reduce uncertainty in the PMP estimates, the revised method was used and led to decrease the PMP estimates (201 mm). These values indicated that the PMP obtained from the revised method and physical approach are closer to the $(P_{24})_{max}$. Because the ratio of the point $PMP_{24}$ to the $(P_{24})_{max}$ in the standard method was high at the study stations, this method is not recommended in this basin.

Due to considering the physical characteristics of the air mass in the hydro-meteorological approach, it is suggested that this approach is used by disregarding uncertainty. If the results of the standard method are used for designing local structures, the construction costs will be unnecessarily high. By including PMP analysis together with extreme rainfall return periods optimum decisions can be made easier. Such studies are crucial for basins with high population and exposed to various kinds of water-related natural disasters.

**5 Acknowledgements**

The authors wish to express their sincere thanks to Mr. Farzad Mahdikhani for invaluable support during the designing of PMP Calculator application.

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

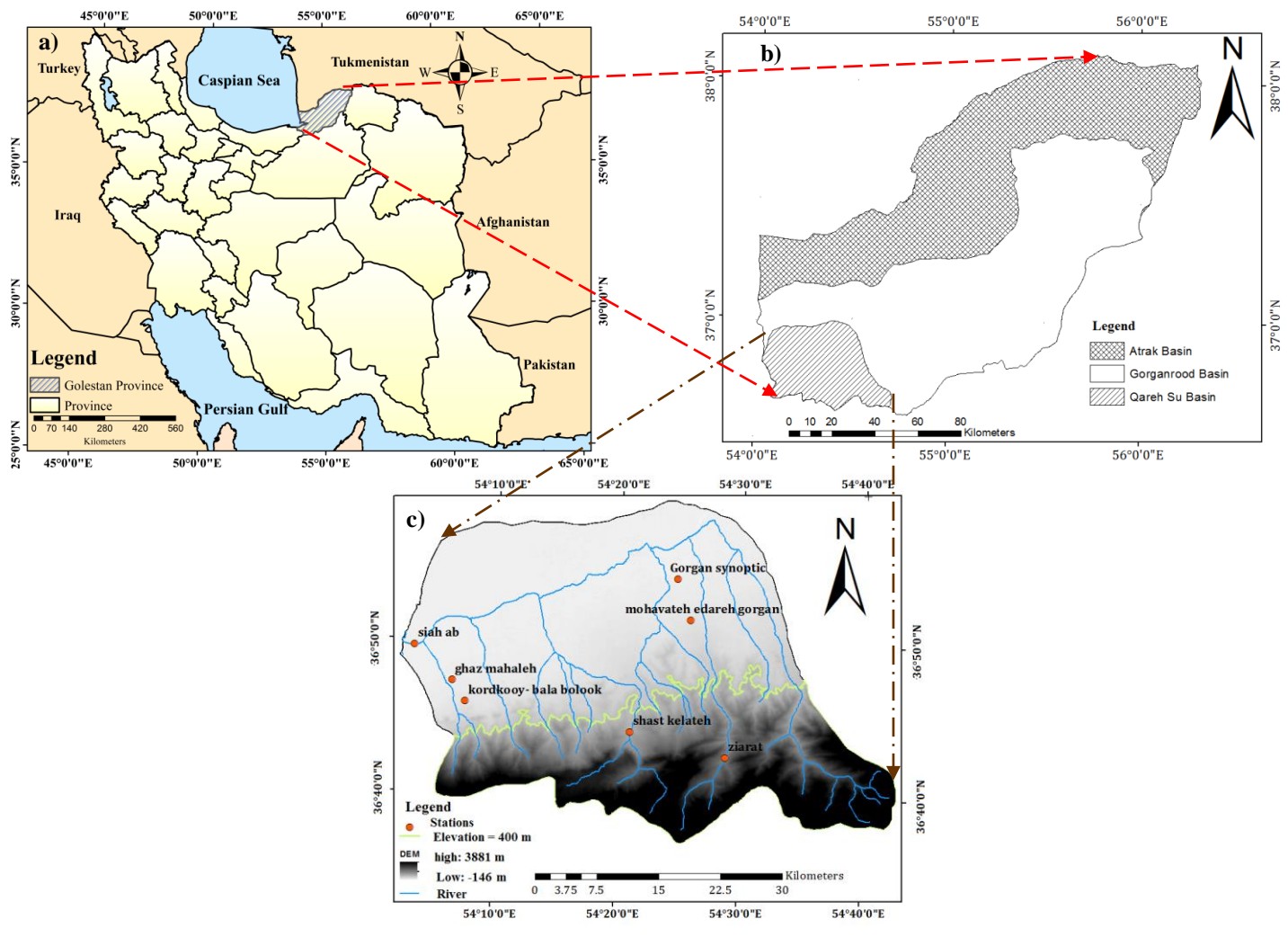

**Figure 1: (a) Golestan province , north of Iran , (b) three main basin of Golestan, (c) the location of stations in the Qareh Su basin in southwestern Golestan**

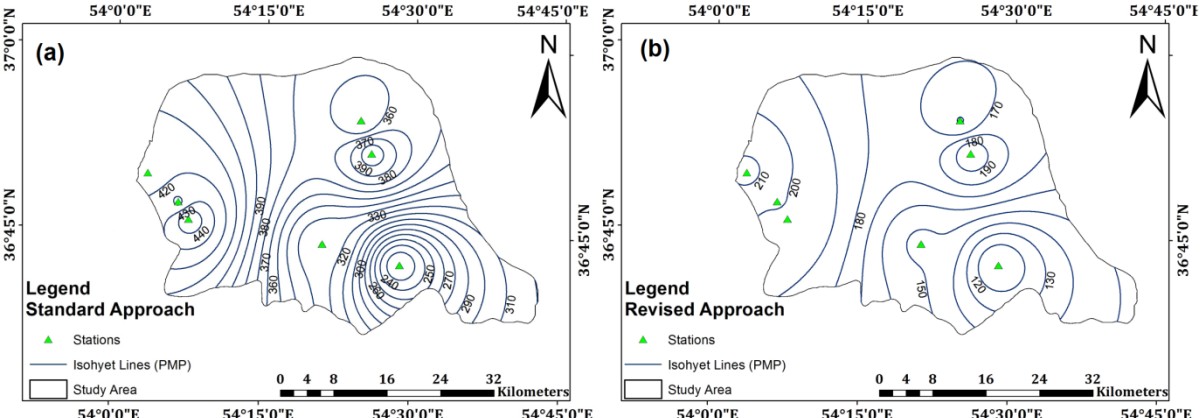

**Figure 2:** The spatial distribution of PMP$_{24}$ using (a) the standard and (b) the revised approaches in study area.

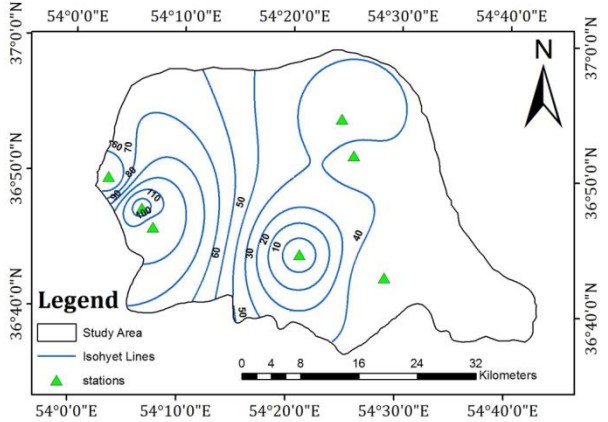

**Figure 3:** Spatial distribution of rainfall for the storm of September 2008 in study area.

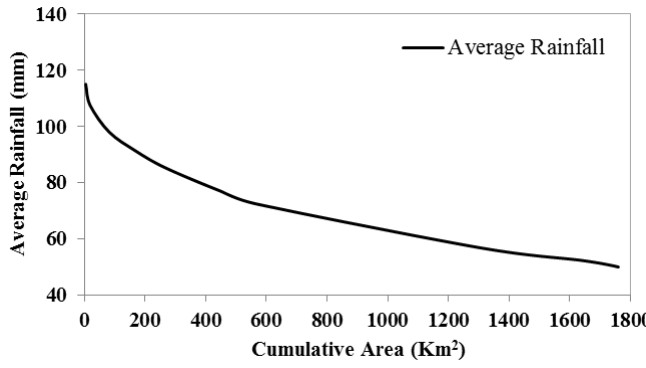

**Figure 4:** Depth-Area-Duration curve for the storm of September 2008 in study area.

