# Peer review of "Probable Maximum Precipitation Estimation in a Humid Climate"

_Natural Hazards and Earth System Sciences, 2018_

## Short Comment (SC1) · 23 Feb 2018

The paper uses standard methods. Publication of the results could be useful for practical application in this region. The authors briefly describe the development of a Java-based software application that could be used to facilitate estimates. It would be helpful if the authors provided information on how a practitioner could access this Java program.

---

## Short Comment (SC2) · 23 Feb 2018

ØšÙĞØśØğ Øğ: To help the reader more quickly understand the analysis, it's better that mentions values of both Km and Kenvelope in the table and a plot of the envelope you derived. I'd also like to see values of the mean and standard deviation of the annual max series for 24 hour precip in the table, not just the CV. Lastly; it would be very helpful to illustrate the steps of the calculations for 1 row of Table 4 in an Appendix.

---

## Author Comment (AC1) · 23 Feb 2018

This paper provides engineering analysis of PMP in a northern region of Iran for which there are no existing values. The study area is located in one of the most flood-prone areas that floods cause billions in damages every year. Qareh Su basin is a part of Golestan province that has a long history of severe damage from the many people dying in floods. Due to the application and importance of PMP in designing structures as input data for calculation of PMF, and lack of comprehensive studies on PMP in the country's watersheds, it is crucial to perform PMP assessment using appropriate methods, such as Physical and statistical methods. For this purpose, two methods of PMP estimation were considered. In order to calculate statistical PMP, a Java-based software application was developed that could be used to facilitate estimates. By using

this software the PMP values in 5-minutes, 1, 6, and 24-hours duration will update every year within the shortest possible time. By using this application, we can determine the best frequency factor based on observed rainfall in each area.

———————————————————

---

## Author Comment (AC2) · 23 Feb 2018

According to (WMO, 2009, page 66, section 4.2) "The greatest value of Km computed from the data for all stations was 15. It was first thought that Km was independent of rainfall magnitude, but it was later found to vary inversely with rainfall: the value of 15 may be too high for areas of generally heavy rainfall and too low for arid areas." Because of the study area is a wet area, the value of Km for wet areas is too high, and therefore revised approach was used to obtain the appropriate value of Km. In order to calculate the Km, the equation 2 was used. Then the maximum value of Km was considered as Km-envelope and was used to the calculation of PMP24. The Km values in standard approach were obtained from Km curves (WMO, 2009; Hershfield, 1965). These curves obtained from 2700 stations over America, while in revised

approach, frequency factor was obtained from observed rainfall over the study area and stations. The frequency factor in revised approach is more reasonable, for it was obtained based on real occurred rainfall over the study area and the result of corresponding PMP is closer to real occurred rainfall over the study area. Reduction of Km in revised approach is not a reason to refuse standard approach; this shows that the standard approach estimates the PMP with more caution while estimating the appropriate value of Km is leading to decreasing the cost of structures that affected by PMP. The results of both approaches and corresponding values of adjustment coefficients are mentioned in attached tables.

**Table I. Required steps in calculation of PMP by standard approach of Hershfield method**

| Stations | Aiah Ab | Kord Kooy | Ziarat | Ghaz Mahalleh | Shast Kelateh | Edareh Gorgan | Gorgan |
|---|---|---|---|---|---|---|---|
| $\overline{x}_n$ | 53.6 | 59.9 | 36.2 | 54.4 | 51.3 | 47.3 | 50.9 |
| $K_m$ = frequency factor | 17.2 | 17.0 | 18.0 | 17.2 | 17.3 | 17.5 | 17.3 |
| $S_n$ | 25.6 | 21.8 | 11.9 | 23.4 | 15.2 | 24.2 | 17.2 |
| Max = Max value of annual series | 150.2 | 104.7 | 63.5 | 132.0 | 92.0 | 139.0 | 95.0 |
| $\overline{x}_{n-m}$ | 50.6 | 58.5 | 34.5 | 52.0 | 50.0 | 44.4 | 49.5 |
| $S_{n-m}$ | 19.1 | 20.6 | 9.8 | 19.1 | 13.6 | 18.0 | 15.5 |
| $\overline{x}_{n-m}/\overline{x}_n$ | 0.944 | 0.977 | 0.951 | 0.955 | 0.975 | 0.939 | 0.973 |
| $\overline{S}_{n-m}/\overline{S}_n$ | 0.747 | 0.944 | 0.831 | 0.817 | 0.891 | 0.744 | 0.902 |
| $C_1$ =Adjustment of $X_n$ for maximum observed event | 0.961 | 0.994 | 0.969 | 0.973 | 0.993 | 0.956 | 0.991 |
| $C_2$ = Adjustment of $S_n$ for maximum observed event | 0.808 | 1.023 | 0.900 | 0.884 | 0.965 | 0.804 | 0.977 |
| $C_3$ =Adjustment of $X_n$ for sample size | 1.003 | 0.996 | 0.996 | 1.002 | 1.002 | 1.004 | 0.996 |
| $C_4$ = Adjustment of $S_n$ for sample size | 1.027 | 1.027 | 1.027 | 1.027 | 1.027 | 1.027 | 1.027 |
| Adjusted Mean= $\overline{x}_n \times C_1 \times C_3$ | 51.7 | 59.3 | 34.9 | 53.0 | 51.0 | 45.4 | 50.2 |
| Adjusted $S_n = S_n \times C_2 \times C_4$ | 21.2 | 22.9 | 10.9 | 21.3 | 15.1 | 20.0 | 17.3 |
| $PMP_1$= Adjusted Mean + $K_m$ ×Adjusted $S_n$ | 416.9 | 447.0 | 232.1 | 418.5 | 312.2 | 394.7 | 349.5 |
| $C_5$ = Adjustment for fixed observational time intervals | 1.13 | 1.13 | 1.13 | 1.13 | 1.13 | 1.13 | 1.13 |
| $PMP_{Point}$ =$PMP_1 \times C_5$ | 471.1 | 505.2 | 562.2 | 472.9 | 352.8 | 446.1 | 394.9 |
| $C_6$ = Adjustment for area-reduction | 0.7 | 0.7 | 0.7 | 0.7 | 0.7 | 0.7 | 0.7 |
| $PMP_{Areal}$ =$PMP_{Point} \times C_6$ | 329.8 | 353.6 | 183.6 | 331.0 | 246.9 | 312.2 | 276.5 |

**Fig. 1.**
Table II. Required steps in calculation of PMP by revised approach of Hershfield method

| Stations | Aiah Ab | Kord Kooy | Ziarat | Ghaz Mahalleh | Shast Kelateh | Edareh Gorgan | Gorgan |
|---|---|---|---|---|---|---|---|
| $\overline{x}_n$ | 53.6 | 59.9 | 36.2 | 54.4 | 51.3 | 47.3 | 50.9 |
| $S_n$ | 25.6 | 21.8 | 11.9 | 23.4 | 15.2 | 24.2 | 17.2 |
| Max | 150.2 | 104.7 | 63.5 | 132.0 | 92.0 | 139.0 | 95.0 |
| Max=Xm | 150.2 | 104.7 | 63.5 | 132.0 | 92.0 | 139.0 | 95.0 |
| $\overline{x}_{n-m}$ | 50.6 | 58.5 | 34.5 | 52.0 | 50.0 | 44.4 | 49.5 |
| $S_{n-m}$ | 19.1 | 20.6 | 9.8 | 19.1 | 13.6 | 18.0 | 15.5 |
| $K_m$ | 5.216 | 2.248 | 2.949 | 4.182 | 3.101 | 5.264 | 2.932 |
| $K_m^* = K_{m-envelope}$ | 5.260 | 5.260 | 5.260 | 5.260 | 5.260 | 5.260 | 5.260 |
| $PMP = \overline{x}_n + K_m^* \times S_n$ | 188.122 | 174.396 | 98.551 | 177.606 | 131.268 | 174.391 | 141.358 |
| $C_1$ = Adjustment for fixed observational time intervals | 1.130 | 1.130 | 1.130 | 1.130 | 1.130 | 1.130 | 1.130 |
| $PMP_{Point} = PMP \times C_1$ | 212.6 | 197.1 | 111.4 | 200.7 | 148.3 | 197.1 | 159.7 |
| $C_2$ = Adjustment for area-reduction | 0.73 | 0.73 | 0.73 | 0.73 | 0.73 | 0.73 | 0.73 |
| $PMP_{Areal} = PMP_{Point} \times C_2$ | 155.18 | 143.86 | 81.29 | 146.51 | 108.28 | 143.85 | 116.61 |

**Fig. 2.**

---

## Referee Comment (RC1) · DR. Sorooshian (Referee) · 19 Mar 2018

**Probable Maximum Precipitation Estimation in a Humid Climate**

The paper focuses on estimating and comparing Probable Maximum Precipitation(PMP) values from different statistical and physical methods. The statistical methods considered were Hershfield and Modified Hershfield methods, and the physical method used was the convergence method.

The manuscript is interesting, and it deserves publication in the NHESS journal. However, the authors need to do a major revision to address some issues and to improve   the paper in terms of organization, flow, content, and grammar.

*Major comments:*

1) The main paragraph in the introduction is considerably long and not clear about its message. It is also weak in terms of flow.   Furthermore, the authors list a large number of studies; however, the strengths, deficiencies, and implications of the cited references to your work are not mentioned and how these findings are relevant to your work. The review of the literature should be presented in a way that the readers can understand what has been done related to the topic in the past and build the argument why your contribution is a valuable extension of the previous work. A one-line summary that may not be even relevant to your approach is not sufficient.

2) the overall goal of the study is not well defined. I suggest considering the following items in introducing the goals of this study:
   a. Do you claim that PMP calculations are not available in that region? Or you think that the current estimates need to be revisited?
   b. Furthermore, explain why you are estimating the PMP24 from both statistical and physical methods?
   c. Do you intend to compare the results obtained from the two and specify which is the better method? In any case, the authors should make their intentions of the study clearer.
   d. The authors also need to describe the statistical metric(s) and measure(s) which should be employed to identify the superior method.

3) The methodology section is very brief. More detailed explanation of the methods and equations are required in order to allow the reproducibility of the implemented approaches. Furthermore, the purpose of some of the equations and calculations is not described and the reader could not understand how they contribute to the overall estimation approach. The general flow of the methodology section also needs to be improved.

4)  The results section must highlight the main findings from each figure and table.

*Minor comments:*

P3L2: This is more suitable for the beginning of the introduction.

P3L11: How is that basin important? Is it important in terms of water supply? Or it has geopolitical importance?

Figure 1: What are the "+" signs in the map? It should be mentioned in the legend. Also, Include the map of Iran, in a larger regional context, in the corner of this figure and demonstrate the location of this basin. Additionally, mention the elevation unit beside "DEM". The "d" letter in the legend is overlapped with the basin boundaries.

P3L13: For which period? Last 30 years?

P3L14: Air pressure? Vapor pressure? Saturated vapor pressure?

P2L13: Are these climatological data taken from the only synoptic station available in your study? If so, please mention it.

P3L14: The sampling frequency and the calculation time-steps should be mentioned. For instance, whether the stations provide hourly values? Or daily? Or for the wind speed data, in what elevation is the wind speed measured? 10m or 2m? It would be good to present this information in a table.

Table 1: Also mention the average annual precipitation in each of these stations.

P4L2: Do you mean the "Annual maximum series"? Does it also work with the "Peak Over Threshold" extreme series?

P4L2: What does this frequency factor mean?

P4L3: Are the "Km" values from the chart method based only on the average extreme value and duration? Are the charts similar for the eastern and western US?

P4L5: Do you mean "The United States"?

P4L5: Why did they modify it? What was wrong with the original approach?

P4L2-L5: The sentence is too long. Also needs grammar revisit.

P4L7-L10: It turns out that only the first equation is used! What is the second equation then used for?

P4L7: Is the $X_{max}$, a single value? Is it the grand maximum, or a time series?

P4L12: What are the differences between these methods? Why did you choose the "convergence" method?

P4L18: Did you also consider the discharge data? If so, mentioned it in the data section. If not, how did you estimate the maximum discharge?

P4L21: What is the purpose of doing "Moisture maximization" and "Wind Maximization"? Are they parts of the convergence model? Or they are different PMP calculation methods? From section 2.3 it turns out to be so; however, it seems to be a different PMP estimation method according to P4L23.

P5L1-L14: How are the FM and MW used? It is not clear from the text that why they are calculated?

P5L21: How was this equation calculated? If this is a polynomial function fitted to the point data, it needs to be shown.

P5L26: The application seems to be of limited use for other regions given the fact that information from limited gauges in one basin is considered in its development.

P6L1: What are the summary of findings from Table 2? What are the differences and what are the sources?

P6L1: Km values and PMP24 values from the standard and modified approaches are considerably different. Which one is more accurate? How is the better approach determined?

P6L2: The isohyetal maps also show significant differences between the PMP24s. How do you discuss and justify this issue?

P6L6: How did you characterize these storms? What measures did you consider in selecting these 8 storms?

Table 3: How many days did each of these storms last?

P6L9: What interpolation method has been used to generate Figure 4?

P6L13: What do you understand from table 4 and table 5?

P7L4: This section is supposed to discuss the physical method. Discussion ion the statistical method should go to section 3.1.

P7L4: For the physical method, is there only one PMP value for the whole basin? Why the physical method gives a different value for each storm, but the statistical method gives one fixed value for the entire period?

P8L14: You compared two statistical methods and the results showed that they lead to considerably different PMP estimates. You did not make any comparison between the different physical methods to show how their results would compare.

P8L15: Why the statistical method gives different PMP values for different locations; however, the physical method gives a single value for the whole basin. It turns out that the PMP values from the statistical method change only in space dimension, but those from the physical method change only in the time dimension.

---

## Referee Comment (RC2) · Anonymous Referee #2 · 22 Mar 2018

[referee-annotated manuscript omitted]

---

## Author Comment (AC3) · 11 Apr 2018

The authors wish to thank the editors and reviewers for their time in their review of our manuscript. We hope that the listed changes have made the manuscript suitable for publication, and we look forward to your response. Response to reviewer was uploaded in PDF format in 2 separate files, too. ——————————————————————————————— ——————————————————————————

Probable Maximum Precipitation Estimation in a Humid Climate The paper focuses on estimating and comparing Probable Maximum Precipitation (PMP) values from different statistical and physical methods. The statistical methods considered were Hershfield and Modified Hershfield methods, and the physical method used was the convergence

method. The manuscript is interesting, and it deserves publication in the NHESS journal. However, the authors need to do a major revision to address some issues and to improve the paper in terms of organization, flow, content, and grammar. The authors wish to thank the editors and reviewers for their time in effort in reviewing our manuscript. We hope the changes listed have made the manuscript suitable for publication and we look forward to your response.

Response to Reviewer: Major comments: (1) The main paragraph in the introduction is considerably long and not clear about its message. It is also weak in terms of flow. Furthermore, the authors list a large number of studies; however, the strengths, deficiencies, and implications of the cited references to your work are not mentioned and how these findings are relevant to your work. The review of the literature should be presented in a way that the readers can understand what has been done related to the topic in the past and build the argument why your contribution is a valuable extension of the previous work. A one-line summary that may not be even relevant to your approach is not sufficient. Re: The text was revised and irrelevant citations were removed. ———————————————————————————————————————————————

2) The overall goal of the study is not well defined. I suggest considering the following items in introducing the goals of this study: a. Do you claim that PMP calculations are not available in that region? Or you think that the current estimates need to be revisited? b. Furthermore, explain why you are estimating the PMP24 from both statistical and physical methods? c. Do you intend to compare the results obtained from the two and specify which is the better method? In any case, the authors should make their intentions of the study clearer. d. The authors also need to describe the statistical metric(s) and measure(s) which should be employed to identify the superior method. Re: a. There are many regions in various parts of the world for which PMP has never been estimated. Qareh Su basin is one of these regions. On the other hand, the accuracy, or reliability, of an estimate, fundamentally depends on the amount and quality

of data available and the depth of analysis. Procedures for estimating PMP cannot be standardized. They vary with the amount and quality of data available, basin size and location, basin and regional topography, storm types producing extreme precipitation, and climate (WMO, 2009). b. In some cases, it is appropriate to make parallel estimates using more than one method, followed by comprehensive analysis in order to acquire reasonable PMP estimates. Therefore, the aim of this study is the estimation of PMP by using two main methods such as statistical methods and physical methods. c. The results of the statistical method are affected by maximum annual 24-h precipitation, while the results of the physical method are affected by dew point temperature, wind speed and direction, air pressure, and precipitation. Because of the results of both method are affected by different factors, comparison of two methods are not investigated. Also, the results of statistical method are point PMP, while physical method provided the areal PMP. Nevertheless, performance criteria were used to a brief comparison. d. Performance criteria as a new section were added to manuscript. The result of them was added to results and discussion. ————————————————————————————————————————————————————

3) The methodology section is very brief. More detailed explanation of the methods and equations are required in order to allow the reproducibility of the implemented approaches. Furthermore, the purpose of some of the equations and calculations is not described and the reader could not understand how they contribute to the overall estimation approach. The general flow of the methodology section also needs to be improved. Re: The required description was added to the text. ————————————————————————————————————————————————————

4) The results section must highlight the main findings from each figure and table. Re: Due to changes in the results and discussion section, we hope the manuscript suitable. ————————————————————————————————————————————————————

Minor comments: P3L2: This is more suitable for the beginning of the introduction. Re:

This sentence was moved to the first section of the manuscript. ——————————
————————————————————————————

P3L11: How is that basin important? Is it important in terms of water supply? Or it has geopolitical importance? Re: Qareh-Su basin is located in Golestan province in the northern parts of Iran with a humid climate. The Qareh-Su basin, with nearly 1760 km2 area is one of the most important basins in the north of Iran. This area is important from the viewpoint of the existence of different cities and villages, population densities, industrial and agricultural centers, flood, and watershed management schemes. 8% of the surface water (equal to 100 million cubic meters) in Golestan province is derived from the Qareh-Su basin. There are two main dams including Kowsar and Shast kalateh to supply water demand of agricultural and residential land located in this area. Also, it is one of the most flood-prone areas that has suffered severe floods throughout its long history, so that in recent years, many people have died in destructive floods. Over the period 1951–2013, the annual average precipitation in this basin is 596 mm. ——————————————————————————
————————————————————

Figure 1: What are the "+" signs in the map? It should be mentioned in the legend. Also, Include the map of Iran, in a larger regional context, in the corner of this figure and demonstrate the location of this basin. Additionally, mention the elevation unit beside "DEM". The "d" letter in the legend is overlapped with the basin boundaries. Re: The "+" signs in the map were used as a grid of ticks. The newer version of Figure 1 already added to the manuscript. ————————————————————————
————————————————————————————

P3L13: For which period? Last 30 years? Re: Over the period 1951–2013, The annual average precipitation in this basin is 596 mm. ——————————————————
—————————————————————————————————

P3L14: Air pressure? Vapor pressure? Saturated vapor pressure? Re: It was air

pressure. It was revised. —————————————————————————————
————————————————————————

P2L13: Are these climatological data taken from the only synoptic station available in your study? If so, please mention it. Re: The required data such as air temperature and rainfall were taken from available climatological, hydrometric (Rain gauge station) and synoptic stations in the study area, but dew point temperature data was taken from an available synoptic station in the study area which is called Gorgan station. —————
————————————————————————————————————

P3L14: The sampling frequency and the calculation time-steps should be mentioned. For instance, whether the stations provide hourly values? Or daily? Or for the wind speed data, in what elevation is the wind speed measured? 10m or 2m? It would be good to present this information in a table. Re: Required information was added to the text. —————————————————————————————————
————————————————

Table 1: Also mention the average annual precipitation in each of these stations. The average annual precipitation in each of these stations was added to the manuscript. —————————————————————————————————
—————————————————

P4L2: Do you mean the "Annual maximum series"? Does it also work with the "Peak Over Threshold" extreme series? Re: It means maximum depth of 24-hour precipitation in each year. —————————————————————————————————
—————————————————

P4L2: What does this frequency factor mean? Re: Km is then the number of standard deviations to be added to obtain PMP. ————————————————————————
————————————————————————————————

P4L3: Are the "Km" values from the chart method based only on the average extreme

value and duration? Are the charts similar for the eastern and western US? Re: According to (WMO, 2009, page 65, Figure 4.1), Km was shown as a function of rainfall duration and mean of annual series (Hershfield, 1965). Yes. ————————————————

————————————————

P4L5: Do you mean "The United States"? Re: Yes. This was revised in manuscript. ————————————————————————————————
————————————————

P4L5: Why did they modify it? What was wrong with the original approach? Re: The original approach was not wrong. It was first thought that Km was independent of rainfall magnitude, but it was later found to vary inversely with rainfall: the value of 15 may be too high for areas of generally heavy rainfall and too low for arid areas." Because of the study area is a wet area, the value of Km for wet areas is too high, and therefore revised approach was used to obtain the appropriate value of Km. In order to calculate the Km, the equation 2 was used. Then the maximum value of Km was considered as Km-envelope and was used to calculation of PMP24. The Km values in standard approach were obtained from Equation 5, based on 24-h Km chart (WMO, 2009; Hershfield, 1965). These curves obtained from 2700 stations over the USA, while in revised approach, frequency factor was obtained from observed rainfall over the study area and stations. The frequency factor in revised approach is more reasonable, for it was obtained based on real occurred rainfall over the study area and the result of corresponding PMP is closer to real occurred rainfall over the study area. Reduction of Km in revised approach is not a reason to refuse standard approach; this shows that the standard approach estimates the PMP with more caution while estimating appropriate value of Km is leading to decrease the cost of structures that affected by PMP. ——————————————————————————————
——————————————————————

P4L2-L5: The sentence is too long. Also needs grammar revisit. Re: The sentence was revised. Km is frequency factor as a function of duration and average of annual

maximum rainfall (the maximum depth of 24-hour precipitation in each year). In other words, Km is then the number of standard deviations to be added to obtain PMP. In this approach, Km is calculated by Km charts which were extracted based on records of rainfall from around 2700 stations in the climatological observation of the United States of America (WMO, 2009). ————————————————————————————————
————————————————————————————

P4L7-L10: It turns out that only the first equation is used! What is the second equation then used for? Re: The second approach is based on the first approach theory. The main difference between these approaches is Km. in the first approach; Km was obtained from the empirical chart, while in the second approach Km is obtained from the actual rainfall in each station and considers the maximum value of Km as a regional value of Km for all stations. ———————————————————————————
————————————————————————————

P4L7: Is the Xmax, a single value? Is it the grand maximum, or a time series? Re: Xmax is a single value for each station and it is maximum depth of rainfall in period of 1951-2014. ————————————————————————————————————
—————————————————

P4L12: What are the differences between these methods? Why did you choose the "convergence" method? Re: The best and the most reliable procedure to estimate the PMP is usually the physical method, which is divided into two procedures, i.e., the orographic and convergence models. The convergence model is used for PMP estimate in Mid-Latitude regions. It is based on the physical characteristics of storms. In convergence model storm physical features such as moist and warm air and movement of moist and warm air, on the basis of dew point temperature and wind speed and wind direction of any storm should be considered. —————————————————————
————————————————————————————————————

P4L18: Did you also consider the discharge data? If so, mentioned it in the data

section. If not, how did you estimate the maximum discharge? Re: Yes. Maximum 24-hours rainfall data was used to determine the date of occurrence the most severe and widespread storms. Then the maximum daily and instant discharge data were used to ensure the date of occurrence storms by comparing Maximum 24-hours rainfall data and maximum daily and instant discharge data. Because of discharge data and rainfall data have a close correlation. ———————————————————————————————

P4L21: What is the purpose of doing "Moisture maximization" and "Wind Maximization"? Are they parts of the convergence model? Or they are different PMP calculation methods? From section 2.3 it turns out to be so; however, it seems to be a different PMP estimation method according to P4L23. Re: You are right. This is typo mistake. It was revised as "The storm maximization factor is calculated by the moisture maximization factor multiplied by wind maximization factor. The moisture maximization method is one of the acceptable procedures to maximize the rainfall values associated with severe storms (Rakhecha and singh, 2009)." ———————————————————————————————

P5L1-L14: How are the FM and MW used? It is not clear from the text that why they are calculated? Re: Finally, PMP id determined by the precipitation depth R (found using DAD curves) multiplied by moisture maximization and wind maximization factors based on Eq. (5). PMP=FM*MW*R ———————————————————————————————

P5L21: How was this equation calculated? If this is a polynomial function fitted to the point data, it needs to be shown. Re: In this study, the equation of each curve was extracted based on R2. Extracted equations are mentioned below: (Fig 1, Fig 2 and Table 1 attached) ———————————————————————————————

P5L26: The application seems to be of limited use for other regions given the fact that

information from limited gauges in one basin is considered in its development. Re: This application can be used in each region. It can calculate PMP by the standard and revised approaches in each region without any limitation. ——————————————
————————————————————————————————————

P6L1: What are the summary of findings from Table 2? What are the differences and what are the sources? Re: Required description was added to the text in line 14-18 (Page 6). ————————————————————————————————————————
————————————————

P6L1: Km values and PMP24 values from the standard and modified approaches are considerably different. Which one is more accurate? How is the better approach determined? Re: Required description was added to the text in line 1-2 and 5-6 (Page 7). Even based on performance criteria including MAE, MSE, RMSE, MAPE, R, and R2, physical method is more accurate than statistical method and revised approach is better than standard approach. Corresponding values of these performance criteria are mentioned below: (Table 2 attached) ——————————————————————————————————
————————————————————————————————————

P6L2: The isohyetal maps also show significant differences between the PMP24s. How do you discuss and justify this issue? Re: Spatial distribution of PMP in standard approach is affected by Km values. After modification of Km by using the maximum of 24 hours precipitation, the spatial distribution of PMP was drawn. ————————————
————————————————————————————————————

P6L6: How did you characterize these storms? What measures did you consider in selecting these 8 storms? Re: First observed rainfall data during 1981-2013 was sorted descending and all observed rainfalls that have the higher depth than Mean 24-hour precipitation was selected. Then the top 8 observed rainfalls are selected so that the Mean 24-hour precipitation depths of 24-hour rainfall in all stations are higher than Mean 24-hour precipitation. Then the date of each storm was checked by maximum

daily and instant discharge data. It should be noted that the criteria used for selection of the rainstorms are mostly based on the severity of the storms. ———————————
————————————————————————————————————————————

Table 3: How many days did each of these storms last? Re: Due to the aim of study that is the calculation of 24-hour PMP, the duration of all storms is 24-hours. ——————
————————————————————————————————————————————

P6L9: What interpolation method has been used to generate Figure 4? Re: In order to show the spatial distribution of precipitation, the precipitation gradient versus elevation was investigated. In each storm that the gradient of precipitation versus the elevation was significant, the isohyet maps were drawn in ArcGIS software using the Digital Elevation Model (DEM) and the gradient of the precipitation equation, otherwise, the IDW (Inverse Distance Weighted) method would be used. In DAD curves, the areas bounded by isohyet lines were calculated by GIS. ——————————————————
—————————————————————————————————————

P6L13: What do you understand from table 4 and table 5? Re: The steps of physical PMP estimation are mentioned in table 4 & 5. Dew point was considered as the moisture inflow into storms. Therefore, Maximum persisting 12-hr dew point was used to calculate the moisture maximization factor. Maximum persisting 12-hr wind speed was used to approximate wind maximization factor and precipitation efficiency. Then the storm maximization factor was calculated by using the moisture maximization factor and wind maximization factor in table 4. In table 5, physical PMP was calculated by using average rainfall and storm maximization factor (PMP Factor) for each storm. ————————————————————————————————————
——————————————————

P7L4: This section is supposed to discuss the physical method. Discussion ion the statistical method should go to section 3.1. Re: Discussions were transferred to section 3.1. ————————————————————————————————————————

————————

P7L4: For the physical method, is there only one PMP value for the whole basin? Why the physical method gives a different value for each storm, but the statistical method gives one fixed value for the entire period? Re: Yes, the result of the physical method is areal PMP (this is an average PMP value for the study area), while the results of statistical methods are point PMP (a particular value for each station). In physical method, each storm is representative of the severest storm that is possible leads to PMP. ———————————————————————————————

————————

P8L14: You compared two statistical methods and the results showed that they lead to considerably different PMP estimates. You did not make any comparison between the different physical methods to show how their results would compare. Re: There are two main methods to estimate PMP including physical method & statistical method. Among all statistical methods, Hershfield method is more common and convenient that has two standards and modified approaches. Manual of WMO has recommended this method for PMP values in particular regions that long-term rainfall data are available (WMO). Also, the best and the most reliable procedure to estimate the PMP is usually the physical method that was investigated in this study. It is necessary mentioned that physical method (convergence model) was selected based on geographical characteristics of the study area which is more applicable in this area. ————————————————

————————————————————————————————

P8L15: Why the statistical method gives different PMP values for different locations; however, the physical method gives a single value for the whole basin. It turns out that the PMP values from the statistical method change only in space dimension, but those from the physical method change only in the time dimension. Re: The results of statistical method are based on rainfall in each station that is related to mean and standard deviation 24-hour rainfall, and Km. So that the result of the statistical method for each station in a basin is different. While the results of physical method are based

on physical characteristics of storm that effect on an extensive area. ——————
——————————————————————————————————————————

Thank you again for your time and effort and for helping us to improve the manuscript.

Please also note the supplement to this comment:
https://www.nat-hazards-earth-syst-sci-discuss.net/nhess-2018-38/nhess-2018-38-
AC3-supplement.zip

————————————————————
2018-38, 2018.

[Figure]

$K_m$ charts that were extracted by authors

**Fig. 1.** Fig 1

[Figure]

**Km charts (WMO, 2009)**

$K_m$ charts (WMO, 2009)

**Fig. 2.** Fig 2

**Equations of frequency factor (K$_m$) that were extracted by authors**

| Duration | Equation | $(R^2)$ |
|---|---|---|
| 5-Minutes | $K_m = -0.0008 \times x^3 + 0.0414 \times x^2 - 0.8951 x + 19.214$ | 0.9896 |
| 1-Hour | $K_m = -5 \times 10^{-6} \times x^3 + 0.0017 \times x^2 - 0.2744 x + 19.825$ | 0.9987 |
| 6-Hour | $K_m = -4 \times 10^{-7} \times x^3 + 0.0003 \times x^2 - 0.1029 x + 19.172$ | 0.9984 |
| 24-Hour | $K_m = -5 \times 10^{-8} x^3 + 8 \times 10^{-5} x^2 - 0.052 x + 19.794$ | 0.9998 |

**Fig. 3.** Table 1

Statistical comparison between $(P_{24})_{max}$ and estimated $PMP_{24}$ values

| method | MAE | MSE | RMSE | MAPE | R(XY) | $R^2$ |
|---|---|---|---|---|---|---|
| Standard | 258.2 | 69090.5 | 262.9 | 241.7 | 0.8 | 0.63 |
| Revised | 64.36 | 4311 | 65.7 | 61.2 | 0.9 | 0.86 |
| Physical | 7.1 | 50.4 | 7.1 | 4.7 | - | - |

**Fig. 4.** Table 2

---

## Author Comment (AC4) · 11 Apr 2018

Probable Maximum Precipitation Estimation in a Humid Climate The authors wish to thank the editors and reviewers for their time in effort in reviewing our manuscript. We hope the changes listed have made the manuscript suitable for publication and we look forward to your response.

———————————————————————————————————- Response to Reviewer: P1L12: At first, define the variable and then use the abbreviation (e.g. frequency factor; Km). Re: Required description was added to the text. ——————————————————————————————————————- P2L14-17: Too many citations..., without commenting their research Improve the syntax of the

sentence. Re: This sentence was corrected. ———————————————————
———————————————————————————- P3L16: ... of 33 years ranging from
... Re: It was corrected. ———————————————————————————
——————————————————- P4L4&5: Improve the syntax of the sentence. Re: It was
corrected. ———————————————————————————————
—————————- P5L22: Previously, you have mentioned that Km is replaced by Kenvelope
value. Now you use equation 5.Please clarify this point. Re: It was first thought that
Km was independent of rainfall magnitude, but it was later found to vary inversely with
rainfall: the value of 15 may be too high for areas of generally heavy rainfall and too low
for arid areas." Because of the study area is a wet area, the value of Km for wet areas
is too high, and therefore revised approach was used to obtain the appropriate value
of Km. In order to calculate the Km, the equation 2 was used. Then the maximum
value of Km was considered as Km-envelope and was used to calculation of PMP24.
The Km values in standard approach were obtained from Equation 5, based on 24-h
Km chart (WMO, 2009; Hershfield, 1965). These curves obtained from 2700 stations
over the USA, while in revised approach, frequency factor was obtained from observed
rainfall over the study area and stations. The frequency factor in revised approach is
more reasonable, for it was obtained based on real occurred rainfall over the study
area and the result of corresponding PMP is closer to real occurred rainfall over the
study area. Reduction of Km in revised approach is not a reason to refuse standard
approach; this shows that the standard approach estimates the PMP with more caution
while estimating appropriate value of Km is leading to decrease the cost of structures
that affected by PMP. ———————————————————————————
——————————————————- P6L2: Discuss the differences between the two approaches.
Re: The second approach is based on the first approach theory. The main difference
between these approaches is Km. in the first approach; Km was obtained from the
empirical chart, while in the second approach Km is obtained from the actual rainfall
in each station and considers the maximum value of Km as a regional value of Km
for all stations. ———————————————————————————

———————————- P6Section3-2: The authors should provide the Spatial distribution of rainfall PMP24 based on physical method, as they have done regarding the other two statistical procedures. Re: The spatial distribution of PMP24 based on physical method was followed by the Spatial distribution of storm that occurred at 10/29/1993. Also, physical PMP result is an average depth for basin. Figure shows the spatial distribution of storm 10/29/1993. (Fig 1 attached.)

————————————————————————————————————————

—- P8L1: Improve the syntax of the sentence. Re: The sentence was revised.

————————————————————————————————————————

—- P8L1-6: This a repetition found also in section "Material and Methods" Re: The sentence was revised. ———————————————————————————————
———————————————- P8L23: The authors should provide statistical metrics such as R2, RMSE, MAE and probability of detection (POD), false alarm ratio (FAR) and critical success index (CSI). These metrics are important to verify the results obtained by the two applied procedures. Re: Common criteria for rainfall such as (MAE, MSE, RMSE, MAPE, R(XY), and R2 was added to the text. Other criteria were not used because it was used for radar-based rainfall. Even based on performance criteria including MAD, MSE, RMSE, MAPE, R, and R2, physical method is more accurate than statistical method and revised approach is better than standard approach. Corresponding values of these performance criteria are mentioned in table 1 (attached)

————————————————————————————————————————

Thank you again for your time and effort and for helping us to improve the manuscript.

Please also note the supplement to this comment:
https://www.nat-hazards-earth-syst-sci-discuss.net/nhess-2018-38/nhess-2018-38-AC4-supplement.zip

———————————————————————

**Fig. 1.** Fig 1

Statistical comparison between $(P_{24})_{max}$ and estimated $PMP_{24}$ values

| method | MAE | MSE | RMSE | MAPE | R(XY) | $R^2$ |
|---|---|---|---|---|---|---|
| Standard | 258.2 | 69090.5 | 262.9 | 241.7 | 0.8 | 0.63 |
| Revised | 64.36 | 4311 | 65.7 | 61.2 | 0.9 | 0.86 |
| Physical | 7.1 | 50.4 | 7.1 | 4.7 | - | - |

**Fig. 2.** Table 1

---

## Author Comment (AC5) · 16 Apr 2018

**Probable Maximum Precipitation Estimation in a Humid Climate**

Zahra Afzali Gorouh[1], Bahram Bakhtiari[1], Kourosh Qaderi[1]

[revised manuscript text omitted]

---

## Author Response (AR1)

**Editor Decision: Reconsider after major revisions (further review by editor and referees)** (30 Apr 2018) by Piero Lionello

Comments to the Author:

Dear Authors,

Considering the comments of the reviewers and the online public discussion, your manuscript is returned to you for a major revision and it will be sent again to the reviewers.

You are required to provide

| | |
|---|---|
| 1) A revised version of the manuscript, where all changes are marked | The final version of manuscript was attached. All changes were explained in this file and were determined in the main article by using blue color. |
| 2) An accompanying letter, where you explain point wise all changes that you have made to the text as response to the comments of the reviewers. | All changes were explained in this file and were determined in the main article by using blue color. |
| I anticipate that your replies to the comments of reviewer 1 are not convincing. The revised version of the introduction is not sufficient to responded to the reviewer's request for improving it, I cannot find in the text that you have tentatively uploaded the improved explanation of the overall goals of your study and the main new scientific findings are not sufficiently clear. | The introduction and main goal were rewritten. We hoped the changes have made the introduction suitable. |
| This negative comment applies also to the abstract, where you write that "The results of this study will be helpful for planning, designing, and management of hydraulic structures and water resources projects in the study area", without providing in it clear information to support this claim. | This sentence was mentioned just as a suggestion at the end of the abstract. Therefore it was deleted from the text and the abstract was rewritten. We hoped the changes have made the abstract suitable. |
| I find the uploaded text AC4 where you reply to reviewer 2 hard to be read, as line breaks are not used to separate the different comments. Please, take care to avoid this problem when providing the accompanying letter mentioned above in point 2). | We are sorry for this problem. Responses to reviewers are mentioned in this file. |
| In general, please make sure labels are readable when the figures are reduced to printed size. This is certainly the case for figure 1 included in AC4, that I do not understand whether you mean to insert in the main article or it is just provided as an answer to reviewer 2. | Figure 1 was redrawn. The attached figure in AC4 is just provided as an answer to reviewer 2, and it is not used in the main article. |
| Further add to your paper a large scale map showing the area of study, whose location would, otherwise, be unclear to readers not familiar with the geography of IRAN. | The new version of figure was added to text. |

Looking forward to your revised version.

Piero Lionello

**Probable Maximum Precipitation Estimation in a Humid Climate**

The paper focuses on estimating and comparing Probable Maximum Precipitation (PMP) values from different statistical and physical methods. The statistical methods considered were Hershfield and Modified Hershfield methods, and the physical method used was the convergence method. The manuscript is interesting, and it deserves publication in the NHESS journal. However, the authors need to do a major revision to address some issues and to improve the paper in terms of organization, flow, content, and grammar.

**The authors wish to thank the editors and reviewers for their time in effort in reviewing our manuscript. We hope the changes listed have made the manuscript suitable for publication and we look forward to your response.**

| Response to Reviewer 1: | |
|---|---|
| *Major comments:* | |
| The main paragraph in the introduction is considerably long and not clear about its message. It is also weak in terms of flow. Furthermore, the authors list a large number of studies; however, the strengths, deficiencies, and implications of the cited references to your work are not mentioned and how these findings are relevant to your work. The review of the literature should be presented in a way that the readers can understand what has been done related to the topic in the past and build the argument why your contribution is a valuable extension of the previous work. A one-line summary that may not be even relevant to your approach is not sufficient. | The text was revised and irrelevant citations were removed. |
| 2) The overall goal of the study is not well defined. I suggest considering the following items in introducing the goals of this study:
a. Do you claim that PMP calculations are not available in that region? Or you think that the current estimates need to be revisited?
b. Furthermore, explain why you are estimating the PMP24 from both statistical and physical methods?
c. Do you intend to compare the results obtained from the two and specify which is the better method? In any case, the authors should make their intentions of the study clearer.
d. The authors also need to describe the statistical metric(s) and measure(s) which should be employed to identify the superior method. | a. There are many regions in various parts of the world for which PMP has never been estimated. Qareh Su basin is one of these regions. On the other hand, the accuracy, or reliability, of an estimate, fundamentally depends on the amount and quality of data available and the depth of analysis. Procedures for estimating PMP cannot be standardized. They vary with the amount and quality of data available, basin size and location, basin and regional topography, storm types producing extreme precipitation, and climate (WMO, 2009).
b. In some cases, it is appropriate to make parallel estimates using more than one method, followed by comprehensive analysis in order to acquire reasonable PMP estimates. Therefore, the aim of this study is the estimation of PMP by using two main methods such as statistical methods and physical methods.
c. The results of the statistical method are affected by maximum annual 24-h precipitation, while the results of the physical method are affected by dew point temperature, wind speed and direction, air pressure, and precipitation. Because of the results of both method are affected by different factors, comparison of two methods are not investigated. Also, the results of statistical method are point PMP, while physical method provided the areal PMP. Nevertheless, performance criteria were used to a brief comparison.
d. Performance criteria as a new section were added to manuscript. The result of them was added to results and discussion. |
| 3) The methodology section is very brief. More detailed explanation of the methods and equations are required in order to allow the reproducibility of the implemented approaches. Furthermore, the purpose of some of the equations and calculations is not described and the reader could not understand how they contribute to the overall estimation approach. The general flow of the methodology section also needs to be improved. | The required description was added to the text. |
| 4) The results section must highlight the main findings from each figure and table. | Due to changes in the results and discussion section, we hope the manuscript suitable. |
| *Minor comments:* | |
| P3L2: This is more suitable for the beginning of the introduction. | This sentence was moved to the first section of the manuscript. |
| P3L11: How is that basin important? Is it important in terms of water supply? Or it has geopolitical importance? | Qareh-Su basin is located in Golestan province in the northern parts of Iran with a |

| | |
|---|---|
| | humid climate. The Qareh-Su basin, with nearly 1760 km2 area is one of the most important basins in the north of Iran. This area is important from the viewpoint of the existence of different cities and villages, population densities, industrial and agricultural centers, flood, and watershed management schemes. 8% of the surface water (equal to 100 million cubic meters) in Golestan province is derived from the Qareh-Su basin. There are two main dams including Kowsar and Shast kalateh to supply water demand of agricultural and residential land located in this area. Also, it is one of the most flood-prone areas that has suffered severe floods throughout its long history, so that in recent years, many people have died in destructive floods. Over the period 1951–2013, the annual average precipitation in this basin is 596 mm. |
| Figure 1: What are the "+" signs in the map? It should be mentioned in the legend. Also, Include the map of Iran, in a larger regional context, in the corner of this figure and demonstrate the location of this basin. Additionally, mention the elevation unit beside "DEM". The "d" letter in the legend is overlapped with the basin boundaries. | The "+" signs in the map were used as a grid of ticks. The newer version of Figure 1 already added to the manuscript. |
| P3L13: For which period? Last 30 years? | Over the period 1951–2013, The annual average precipitation in this basin is 596 mm. |
| P3L14: Air pressure? Vapor pressure? Saturated vapor pressure? | It was air pressure. It was revised. |
| P2L13: Are these climatological data taken from the only synoptic station available in your study? If so, please mention it. | The required data such as air temperature and rainfall were taken from available climatological, hydrometric (Rain gauge station) and synoptic stations in the study area, but dew point temperature data was taken from an available synoptic station in the study area which is called Gorgan station. |
| P3L14: The sampling frequency and the calculation time-steps should be mentioned. For instance, whether the stations provide hourly values? Or daily? Or for the wind speed data, in what elevation is the wind speed measured? 10m or 2m? It would be good to present this information in a table. | Required information was added to the text. |
| Table 1: Also mention the average annual precipitation in each of these stations. | The average annual precipitation in each of these stations was added to the manuscript. |
| P4L2: Do you mean the "Annual maximum series"? Does it also work with the "Peak Over Threshold" extreme series? | It means maximum depth of 24-hour precipitation in each year. |
| P4L2: What does this frequency factor mean? | $K_m$ is then the number of standard deviations to be added to obtain PMP. |
| P4L3: Are the "Km" values from the chart method based only on the average extreme value and duration? Are the charts similar for the eastern and western US? | According to (WMO, 2009, page 65, Figure 4.1), Km was shown as a function of rainfall duration and mean of annual series (Hershfield, 1965). Yes. |
| P4L5: Do you mean "The United States"? | Yes. This was revised in manuscript. |
| P4L5: Why did they modify it? What was wrong with the original approach? | The original approach was not wrong. It was first thought that $K_m$ was independent of rainfall magnitude, but it was later found to vary inversely with rainfall: the value of 15 may be too high for areas of generally heavy rainfall and too low for arid areas." Because of the study area is a wet area, the value of $K_m$ for wet areas is too high, and therefore revised approach was used to obtain the appropriate value of $K_m$. In order to calculate the $K_m$, the equation 2 was used. Then the maximum value of $K_m$ was considered as $K_{m-envelope}$ and was used to calculation of $PMP_{24}$. The $K_m$ values in standard approach were obtained from Equation 5, based on 24-h $K_m$ chart (WMO, 2009; Hershfield, 1965). These curves obtained from 2700 stations over the USA, while in revised approach, frequency factor was obtained from observed rainfall over the study area and stations. The frequency factor in revised approach is more reasonable, for it was obtained based on real occurred rainfall over the study area and the result of corresponding PMP is closer to real occurred rainfall over the study area. Reduction of $K_m$ in revised approach is not a reason to refuse standard approach; this shows that the standard approach estimates the PMP with more caution while estimating appropriate value of $K_m$ is leading to decrease the cost of structures that affected by PMP. |
| P4L2-L5: The sentence is too long. Also needs grammar revisit. | The sentence was revised. |

| | |
|---|---|
| | $K_m$ is frequency factor as a function of duration and average of annual maximum rainfall (the maximum depth of 24-hour precipitation in each year). In other words, $K_m$ is then the number of standard deviations to be added to obtain PMP. In this approach, $K_m$ is calculated by $K_m$ charts which were extracted based on records of rainfall from around 2700 stations in the climatological observation of the United States of America (WMO, 2009). |
| P4L7-L10: It turns out that only the first equation is used! What is the second equation then used for? | The second approach is based on the first approach theory. The main difference between these approaches is $K_m$. in the first approach; $K_m$ was obtained from the empirical chart, while in the second approach $K_m$ is obtained from the actual rainfall in each station and considers the maximum value of $K_m$ as a regional value of $K_m$ for all stations. |
| P4L7: Is the Xmax, a single value? Is it the grand maximum, or a time series? | $X_{max}$ is a single value for each station and it is maximum depth of rainfall in period of 1951-2014. |
| P4L12: What are the differences between these methods? Why did you choose the "convergence" method? | The best and the most reliable procedure to estimate the PMP is usually the physical method, which is divided into two procedures, i.e., the orographic and convergence models. The convergence model is used for PMP estimate in Mid-Latitude regions. It is based on the physical characteristics of storms. In convergence model storm physical features such as moist and warm air and movement of moist and warm air, on the basis of dew point temperature and wind speed and wind direction of any storm should be considered. |
| P4L18: Did you also consider the discharge data? If so, mentioned it in the data section. If not, how did you estimate the maximum discharge? | Yes. Maximum 24-hours rainfall data was used to determine the date of occurrence the most severe and widespread storms. Then the maximum daily and instant discharge data were used to ensure the date of occurrence storms by comparing Maximum 24-hours rainfall data and maximum daily and instant discharge data. Because of discharge data and rainfall data have a close correlation. |
| P4L21: What is the purpose of doing "Moisture maximization" and "Wind Maximization"? Are they parts of the convergence model? Or they are different PMP calculation methods? From section 2.3 it turns out to be so; however, it seems to be a different PMP estimation method according to P4L23. | You are right. This is typo mistake. It was revised as "The storm maximization factor is calculated by the moisture maximization factor multiplied by wind maximization factor. The moisture maximization method is one of the acceptable procedures to maximize the rainfall values associated with severe storms (Rakhecha and singh, 2009)." |
| P5L1-L14: How are the FM and MW used? It is not clear from the text that why they are calculated? | Finally, PMP id determined by the precipitation depth R (found using DAD curves) multiplied by moisture maximization and wind maximization factors based on Eq. (5). $$PMP = FM \times MW \times R$$ |
| P5L21: How was this equation calculated? If this is a polynomial function fitted to the point data, it needs to be shown. | In this study, the equation of each curve was extracted based on $R^2$. Extracted equation are mentioned below:  $K_m$ charts that were extracted by authors |

**Km charts (WMO, 2009)**

**Equations of frequency factor (Km) that were extracted by authors**

| Duration | Equation | ($R^2$) |
|---|---|---|
| 5-Minutes | $K_m = -0.0008 \times x^3 + 0.0414 \times x^2 - 0.8951x + 19.214$ | 0.9896 |
| 1-Hour | $K_m = -5 \times 10^{-6} \times x^3 + 0.0017 \times x^2 - 0.2744x + 19.825$ | 0.9987 |
| 6-Hour | $K_m = -4 \times 10^{-7} \times x^3 + 0.0003 \times x^2 - 0.1029x + 19.172$ | 0.9984 |
| 24-Hour | $K_m = -5 \times 10^{-8} x^3 + 8 \times 10^{-5} x^2 - 0.052x + 19.794$ | 0.9998 |

| | |
|---|---|
| P5L26: The application seems to be of limited use for other regions given the fact that information from limited gauges in one basin is considered in its development. | This application can be used in each region. It can calculate PMP by the standard and revised approaches in each region without any limitation. |
| P6L1: What are the summary of findings from Table 2? What are the differences and what are the sources? | Required description was added to the text in line 14-18 (Page 6). |
| P6L1: Km values and PMP24 values from the standard and modified approaches are considerably different. Which one is more accurate? How is the better approach determined? | Required description was added to the text in line 1-2 and 5-6 (Page 7). Even based on performance criteria including MAE, MSE, RMSE, MAPE, r, and $R^2$, physical method is more accurate than statistical method and revised approach is better than standard approach. Corresponding values of these performance criteria are mentioned below: Statistical comparison between $(P_{24})_{max}$ and estimated $PMP_{24}$ values |

| Method | MAE | MSE | RMSE | MAPE | r | $R^2$ |
|---|---|---|---|---|---|---|
| Standard | 258.2 | 69090.5 | 262.9 | 241.7 | 0.8 | 0.63 |
| Revised | 64.36 | 4311 | 65.7 | 61.2 | 0.9 | 0.86 |
| Physical | 7.1 | 50.4 | 7.1 | 4.7 | - | - |

| | |
|---|---|
| P6L2: The isohyetal maps also show significant differences between the PMP24s. How do you discuss and justify this issue? | Spatial distribution of PMP in standard approach is affected by Km values. After modification of Km by using the maximum of 24 hours precipitation, the spatial distribution of PMP was drawn. |
| P6L6: How did you characterize these storms? What measures did you consider in selecting these 8 storms? | First observed rainfall data during 1981-2013 was sorted descending and all observed rainfalls that have the higher depth than Mean 24-hour precipitation was selected. Then the top 8 observed rainfalls are selected so that the Mean 24-hour precipitation depths of 24-hour rainfall in all stations are higher than Mean 24-hour precipitation. Then the date of each storm was checked by maximum daily and instant discharge data. It should be noted that the criteria used for selection of the rainstorms are mostly based on the severity of the storms. |
| Table 3: How many days did each of these storms last? | Due to the aim of study that is the calculation of 24-hour PMP, the duration of all storms is 24-hours. |
| P6L9: What interpolation method has been used to generate Figure 4? | In order to show the spatial distribution of precipitation, the precipitation gradient versus elevation was investigated. In each storm that the gradient of precipitation versus the elevation was significant, the isohyet maps were drawn in ArcGIS software using the Digital Elevation Model (DEM) and the gradient of the precipitation equation, otherwise, the IDW (Inverse Distance Weighted) method would be used. In DAD curves, the areas bounded by isohyet lines were calculated by GIS. |
| P6L13: What do you understand from table 4 and table 5? | The steps of physical PMP estimation are mentioned in table 4 & 5. Dew point was |

| | considered as the moisture inflow into storms. Therefore, Maximum persisting 12-hr dew point was used to calculate the moisture maximization factor. Maximum persisting 12-hr wind speed was used to approximate wind maximization factor and precipitation efficiency. Then the storm maximization factor was calculated by using the moisture maximization factor and wind maximization factor in table 4. In table 5, physical PMP was calculated by using average rainfall and storm maximization factor (PMP Factor) for each storm. |
|---|---|
| P7L4: This section is supposed to discuss the physical method. Discussion ion the statistical method should go to section 3.1. | Discussions were transferred to section 3.1. |
| P7L4: For the physical method, is there only one PMP value for the whole basin? Why the physical method gives a different value for each storm, but the statistical method gives one fixed value for the entire period? | Yes, the result of the physical method is areal PMP (this is an average PMP value for the study area), while the results of statistical methods are point PMP (a particular value for each station). In physical method, each storm is representative of the severest storm that is possible leads to PMP. |
| P8L14: You compared two statistical methods and the results showed that they lead to considerably different PMP estimates. You did not make any comparison between the different physical methods to show how their results would compare. | There are two main methods to estimate PMP including physical method & statistical method. Among all statistical methods, Hershfield method is more common and convenient that has two standards and modified approaches. Manual of WMO has recommended this method for PMP values in particular regions that long-term rainfall data are available (WMO). Also, the best and the most reliable procedure to estimate the PMP is usually the physical method that was investigated in this study. It is necessary mentioned that physical method (convergence model) was selected based on geographical characteristics of the study area which is more applicable in this area. |
| P8L15: Why the statistical method gives different PMP values for different locations; however, the physical method gives a single value for the whole basin. It turns out that the PMP values from the statistical method change only in space dimension, but those from the physical method change only in the time dimension. | The results of statistical method are based on rainfall in each station that is related to mean and standard deviation 24-hour rainfall, and Km. So that the result of statistical method for each station in a basin is different. While the results of physical method are based on physical characteristics of storm that effect on an extensive area. |

**Thank you again for your time and effort and for helping us to improve the manuscript.**

Response to Reviewer:

| Reviewer 2 | |
|---|---|
| P1L12: At first, define the variable and then use the abbreviation (e.g. frequency factor; $K_m$). | Required description was added to the text. |
| P2L14-17: Too many citations..., without commenting their research Improve the syntax of the sentence. | This sentence was corrected. The sentence was corrected. |
| P3L16: ... of 33 years ranging from ... | It was corrected. |
| P4L4&5: Improve the syntax of the sentence. | It was corrected. |
| P5L22: Previously, you have mentioned that Km is replaced by $K_{envelope}$ value. Now you use equation 5.Please clarify this point. | It was first thought that $K_m$ was independent of rainfall magnitude, but it was later found to vary inversely with rainfall: the value of 15 may be too high for areas of generally heavy rainfall and too low for arid areas." Because of the study area is a wet area, the value of $K_m$ for wet areas is too high, and therefore revised approach was used to obtain the appropriate value of $K_m$. In order to calculate the $K_m$, the equation 2 was used. Then the maximum value of $K_m$ was considered as $K_{m\text{-}envelope}$ and was used to calculation of $PMP_{24}$. The $K_m$ values in standard approach were obtained from Equation 5, based on 24-h $K_m$ chart (WMO, 2009; Hershfield, 1965). These curves obtained from 2700 stations over the USA, while in revised approach, frequency factor was obtained from observed rainfall over the study area and stations. The frequency factor in revised approach is more reasonable, for it was obtained based on real occurred rainfall over the study area and the result of corresponding PMP is closer to real occurred rainfall over the study area. Reduction of $K_m$ in revised approach is not a reason to refuse standard approach; this shows that the standard approach estimates the PMP with more caution while estimating appropriate value of $K_m$ is leading to decrease the cost of structures that affected by PMP. |
| P6L2: Discuss the differences between the two approaches. | The second approach is based on the first approach theory. The main difference between these approaches is $K_m$. in the first approach; $K_m$ was obtained from the empirical chart, while in the second approach $K_m$ is obtained from the actual rainfall in each station and considers the maximum value of $K_m$ as a regional value of $K_m$ for all stations. |
| P6Section3-2: The authors should provide the Spatial distribution of rainfall PMP24 based on physical method, as they have done regarding the other two statistical procedures. | The spatial distribution of $PMP_{24}$ based on physical method was followed by the Spatial distribution of storm that occurred at 10/29/1993. Also, physical PMP result is an average depth for basin. Figure shows the spatial distribution of storm 10/29/1993.  |
| P8L1: Improve the syntax of the sentence. | The sentence was revised. |
| P8L1-6: This a repetition found also in section "Material and Methods" | The sentence was revised. |
| P8L23: The authors should provide statistical metrics such as R2, RMSE, MAE and probability of detection (POD), false alarm ratio (FAR) and critical success index (CSI). These metrics are important to verify the results obtained by the two applied procedures. | Common criteria for rainfall such as (MAE, MSE, RMSE, MAPE, r, and $R^2$ was added to the text. Other criteria were not used because it was used for radar-based rainfall. Even based on performance criteria including MAD, MSE, RMSE, MAPE, R, and R2, physical method is more accurate than statistical method and revised approach is better |

than standard approach. Corresponding values of these performance criteria are mentioned below:

Statistical comparison between $(P_{24})_{max}$ and average estimated $PMP_{24}$ values

| Method | MAE | MSE | RMSE | MAPE | r | $R^2$ |
|---|---|---|---|---|---|---|
| Standard | 258.2 | 69090.5 | 262.9 | 241.7 | 0.8 | 0.63 |
| Revised | 64.36 | 4311 | 65.7 | 61.2 | 0.9 | 0.86 |
| Physical | 7.1 | 50.4 | 7.1 | 4.7 | - | - |

**Thank you again for your time and effort and for helping us to improve the manuscript.**

---

## Referee Report (RR1)

**Probable Maximum Precipitation Estimation in a Humid Climate**

By: Afzaligorouh et al.

In general, the authors have addressed many of the concerns and issues that were raised in my previous review of the manuscript. However, the introduction section is still poor in terms of content. A comprehensive review of the relevant literature is lacking. Also, the result's section needs to be updated to provide additional discussion on the results they obtained.

I recommend  **Moderate Revision**, and specifically addressing issues related to the introduction and the results section.

**Major Comments:**

1. Introduction: The introduction is too short, and the body of the introduction section is only one paragraph, lacking sufficient background information. Furthermore, a review of literature on the physical approaches for PMP estimation is lacking which can enhance the introduction.
2. The results section needs to be updated to add additional discussions and analyses on different aspects of the results they already obtained. Just providing one or two sentences that are obvious from the figures or tables is not   appropriate for a paper at this level.
3. An appendix section must be added to give a brief overview of the PMP Calculator app, along with its download link for the interested readers.
4. Authors are strongly encouraged to check the grammar and language of the manuscript before resubmission. Some of these errors are mentioned here, but there are more errors and typos that need to be corrected.

**Minor Comments:**

P1L19: Must be "rainfalls"

P1L19: delete "one" , replace with "among"

P1L19: What are social damages? Damages to societies seems to be more relevant here.

P1L22: Replace "specific project" with "hydrologic infrastructure"

P1L25: Where does the quotation end?

P1L26: "A statistical …" Review of statistical methods should go to a separate paragraph.

P1L29:30 years "of" daily data …

P1L28: "basins"

P1L29: "kilometers"

P1L29: Mention the names of other statistical methods and discuss their differences. Also mention why the Hershfield method is more popular. Review of statistical methods must be a separate paragraph for itself.

P2L1: Any references?

P2L2: Discuss the physical methods in a separate paragraph.

P2L2: "characteristic of the deterministic…" Not sure what you mean!

P2L6: Some physical methods are mentioned; however, nothing special about their characteristics and differences have been mentioned. The only thing mentioned is that they are not easy to use. Please consider adding more details about the different physical methods, and their differences and pros and cons. Also, keep in mind that the difficulty in estimation is not the case in many parts of the world.

P2L8: Comparison of the physical and statistical methods need to go to a separate paragraph. A more detailed review of the literature is required. For instance, why in some regions the two methods give similar results and why in some other regions they are totally different?

P2L11: "The results of these researchers have indicated that although the statistical approaches provide larger estimates of PMP, it is proposed for areas where hourly rainfall, dew point temperature, wind speed, and vertical radiosonde measurements are unavailable"

The first and the second statements are irrelevant to each other.

P2L18-19: It has already been mentioned in L11.

P2L19: What is the overall conclusion from these studies?

P2L20: Replace "was" with "is"

P2L20: delete "written" and write "prepare a" instead

P2L20: Estimation alone is not a good goal for a paper. You can draw more useful information from your results. For instance, using the PMP24 maps, you can specify the regions that are more likely to experience intense storms. Such information could be useful for water resources planning and management, flood risk assessment, and catastrophe management.

P3L2: Where do you want to put figure 1?

> For figure 1, it is also suggested to name each of the small figures, as a, b, and c. Then, give a short description in the figure caption for each of them.

P3L17: Delete "then"

P3L18: To be added to what?

P3L19: Delete "of America"

P4L1: should be "the" standard deviation

P4L2: To do what? Say, the goal first; then, mention the steps.

P4L12: Information about the discharge data is still missing in the data section. Add it!

P5L12: mean "squared" error

P6L20: Provide more discussion on figures. Is there any specific gradient in the PMP values? Which parts of the basin experience more severe storms? Is the basin homogenous in this regard? Which parts of the basin experiences more extreme precipitations? Why?

P8L4: Replace "have been" with "are"

P8L13: delete "s" from "whiles"

P8L14: Not sure what does the most moderate mean!

P9L4: "Based on …." not relevant to the previous sentences. It could be a separate paragraph, joint with the next paragraph.

---

## Referee Report (RR2)

**Probable Maximum Precipitation Estimation in a Humid Climate**

By: Afzaligorouh et al.

The authors have addressed most of the concerns I raised in the previous round of review. My only additional comment is to revise the manuscript from the grammatical point of view. There are still grammatical issues in the manuscript(e.g. Page 3, lines 7-9) that needs to be corrected prior to the final publication.

Kind Regards,

---

## Author Response (AR2)

**Probable Maximum Precipitation Estimation in a Humid Climate**

Dear Editor,

The authors appreciate the editors and reviewers. We checked the manuscript carefully for typos, and co-authors' affiliations, terminology, updates of data in tables, or updates of variables in equations. A point-by-point response to the Editors' and Reviewers' comments is below. Also, all changes were determined in the main text by using red color.

With warm regards,
Corresponding Author,
Bakhtiari, B.

**Comments:**

**Abstract**

| | |
|---|---|
| 1. The information on the frequency factor is not in my view so relevant to be put in the abstract (further you missed to add its units!). I suggest to delete it unless you have a strong argument to keep it (which is not clear in the abstract and should be added to it) | This sentence was deleted from abstract. |
| 2. Considering uncertainty of computed values, is rather clumsy to add so many significant digits to your results in the abstract. Values to be reported are 448, 201 and 143 mm, with no further digits. | Mentioned values and related sentences were revised. |
| 3. I would add a clear statement that the most reliable estimate for PMP is 143mm (if I get it right your conclusions!) | Of course. It's correct. Physical result is the most reliable estimate for PMP which is mentioned in the abstract. |

**Introduction**

| | |
|---|---|
| Local context should be better explained in the introduction. What are the consequences of extreme precipitation in the region? For which specific applications in the considered region is the value of PMP needed? | Required revision was done and was added to text. Extreme rainfalls and flash floods which occurred in the spring and summer seasons are the most common hazards in northern Iran include the southern Caspian region representing provinces of Mazandaran, Gilan, and Golestan. In recent years, Golestan province experienced deadly floods in its historical data. Due to consequences of extreme precipitations and floods in this region, it is necessary to estimate the values of PMP and PMF to reduce the risk of them. The value of PMP is needed for designing irrigation and drainage channels, sewage collection and disposal systems, and the maximum amount of water entering the reservoirs in this region. |
| Minor point: Line 18, page 3 $K_m$ is NOT number, it is a value "In other words, $K_m$ is then the number of standard deviations to be added to obtain PMP." | This sentence is from "Manual on Estimation of Probable Maximum Precipitation, page 66, section 4.2-1, Line 12". If the maximum observed rainfall $X_m$ is substituted for $X_t$, and $K_m$ for $K$, $K_m$ is then the number of standard deviations to be added to  to obtain $X_m$, or $$X_t = \overline{X}_n + K_m S_n \qquad (4.2)$$ |

**Conclusions**

| | |
|---|---|
| *Pleas, insert a discussion of the actual meaning of PMP. From your description, I understand that is it an upper limit that you expected NEVER to be exceeded. However, "NEVER" is quite problematic to be consistently maintained in reality if you wait sufficiently long (eventually millions of years)! | Required revision was done and was added to text. In the theory definition, the PMP refers to the upper bound with a zero probability of exceedance. In practice, these estimates are based on the steps that hydrometeorologists use to maximize observed large storms to achieve PMP value. Therefore, there is a very small probability that the operational estimates of PMP may be exceeded. |
| A short paragraph relating PMP to the return time values of other approaches such as General Extreme Value theory could be useful. Example: if $P\_50$ is the accumulated precipitation in 24 hours that is expected to be reached once every 50years, how are $P\_50m$ and PMP related?

My comment 2 of the abstract applies to the conclusions as well | The sentences associated with the return period and General Extreme Value theory was added. In response to "if $P\_50$ is the accumulated precipitation in 24 hours that is expected to be reached once every 50years, how are $P\_50m$ and PMP related?" it should be noted that the relationship of $P_{50}$ and PMP was investigated in the study area. The results showed that there is a good correlation between $P_{50}$ and PMP in 99% level. Since this matter has not been discussed in this manuscript, related explanations have not been added to the text. If it is necessary, we will add it to the manuscript. Also, the PMP values were revised based on reviewer's comments. |
| * Please, relate the results to an estimate of the actual hazard and its consequences. What would be the local impact of a PMP with a value of 143mm? and if this value were wrong and the higher estimate correct (448mm), what would it be the consequences of this error? Finally, are local structures adequate if PMP is 143mm? and if it is 448 mm? | Required sentences were added to text. It should be noted that all of these approaches have uncertainty in the estimation of PMP. In the statistical approach, significant uncertainty can take place from the use of the enveloping curve of the frequency factor, and uncertainty in the sample mean and standard deviation. Therefore, Hershfield's frequency factor in standard method led to overestimate PMP (448 mm). In order to reduce uncertainty in the PMP estimates, the revised method was used and led to decrease the PMP estimates (201 mm). These values indicated that the PMP obtained from the revised method and physical approach are closer to the $(P_{24})_{max}$. Since the magnitude of point PMP at an individual station should normally not exceed three times the highest observed rainfall from a long period of rainfall data (Hershfield, 1962), the use of the standard method is not recommended in this basin. Because the ratio of point $PMP_{24}$ at the study stations in the standard method to $(P_{24})_{max}$ was more than 3. Due to considering the physical characteristics of the air mass in the hydrometeorological approach, it is suggested that this approach is used by disregarding uncertainty. If the results of the standard method are used for designing local structures, the construction costs will be increased. By including PMP analysis together with extreme rainfall return periods optimum decisions can be made easier. Such studies are crucial for basins with high population and exposed to various kinds of water-related natural disasters. Due to the existence of Kowsar dam in this area, it seems that the amount of precipitation adequate and rational for this dam. |

The authors wish to thank the editors and reviewers for their time in effort in reviewing our manuscript. We hope the changes listed have made the manuscript suitable for publication and we look forward to your response.
Thank you again for your time and effort and for helping us to improve the manuscript.

---

## Author Response (AR3)

**Probable Maximum Precipitation Estimation in a Humid Climate**

| Editor's Comments: | |
|---|---|
| **Comments to the Author:** | |
| Dear Author:

In my email explaining my previous decision, I asked you to consider the comments of reviewer 1 and to provide a revised version with the information and discussion that he/she recommended. I also anticipated that your manuscript was going to be returned to reviewer 1. However, in your "Author's Response", I find only your replies to my comments. Please, provide a version of the manuscript accounting also for the comments of reviewer 1 and your explicit and pointwise responses to them. | Dear Editor,

The authors appreciate the editors and reviewers. Also, we would like to express our much apologies for the lack of reviewers response. The manuscript carefully was checked for typos, and co-authors' affiliations, terminology, updates of data in tables, or updates of variables in equations. A point-by-point response to the Editors' and Reviewers' comments is below. Also, all changes were determined in the main text by using red color.

With warm regards,

Corresponding Author,

Bakhtiari, B. |
| Question: | Author's Response: |
| Further, I am not fully convinced by some replies of yours.

1) The statement "In the theory definition, the PMP refers to the upper bound with a zero probability of exceedance." Is in my view misleading, in general, and should be deleted. If you have a Gaussian distribution, for whatever value of $K\_M$ there will be probability (vanishingly small as $K\_M$ increases) of exceeding the PMP computed using (1). The rest of the sentence is acceptable to me, but it should be slightly rephrased after the deletion. | 1) The sentence was deleted based on the editor's suggestion, and the following sentence has been replaced: "In the theory definition, the PMP is the extreme rainfall for a given duration that is physically possible over an area" (Page 12, Line 2). |
| 2) I think that the information that PMP and $P\_50$ are numerically similar is important and should be added to the manuscript. | 2) Statistical analysis was performed on the $P_{50}$ and the PMP data, based on the editor's suggestion. The results show the significant correlation and are added on (Page 9, Line4). |
| 3) If $P\_50$ and PMP are numerically similar, what is the advantage of using PMP instead of computing $P\_50$ using the Generalized Extreme Value theory? A short comment about this should be added. | 3) The Generalized Extreme Value theory was used to calculate 50-year precipitation ($P_{50}$). For this purpose, the GEV model was fitted on rainfall data. The results showed that there is a significant correlation between the standard and revised estimates of $PMP_{24}$ and $P_{50}$. The values of the coefficient of determination ($R^2$) between the standard and revised estimates of $PMP_{24}$ and $P_{50}$ were 0.97 and 0.98, respectively. The relationships of $PMP_{24}$ and $P_{50}$ are defined by Eq. (13 and 14).

$$PMP_{24}=3.85(P_{50})-24.1 \qquad (13)$$
$$PMP_{24}=1.91(P_{50})-20.3 \qquad (14)$$

The use of Hershfield was recommended, for the application of the GEV model was led to underestimates the upper tail. |
| 4) You write that "Since the magnitude of point PMP at an individual station should normally not exceed three times the highest observed rainfall from a long period of rainfall data (Hershfield, 1962), …".This statement sounds strange to me.
The factor three would imply a huge overestimation, especially if the observation cover along (suppose 100-year long) period. The involved factor should have some dependence with the length of the period covered by the observation and should decrease for long periods. Please, clarify/correct | 4) The sentence was deleted based on the editor's suggestion, and the following sentence has been replaced: "Because the ratio of the point $PMP_{24}$ to the $(P_{24})_{max}$ in the standard method was high at the study stations, this method is not recommended in this basin" and is added on (Page 12, Line 23&24). |
| 5) Instead of "the construction costs will be increased", I would write "the construction costs will be unnecessarily high". A clear conclusion of your work is that in this basin the standard Hershfield Method is not adequate (and you write this).
I would also emphasize in the abstract and in the conclusions that it produces unrealistically large PMP values for construction costs. | The sentence "the construction costs will be increased" was revised to "the construction costs will be unnecessarily high" according to the editor (Page 12, Line 26&27).

The sentence "the standard method gives uselessly large PMP values for construction costs." Was added to abstract (Page 1, Line 22). |
| A further comment: I do not see the relevance of the info provided by figure 2and I suggest to remove it and its reference in the text. | Figure 2 was removed based on this comment. |

**Probable Maximum Precipitation Estimation in a Humid Climate**

| Reviewer's Comments: | |
|---|---|
| **Question:** | **Author's Response:** |
| **Probable Maximum Precipitation Estimation in a Humid Climate**

By: Afzali-Gorouh et al.

In general, the authors have addressed many of the concerns and issues that were raised in my previous review of the manuscript. However, the introduction section is still poor in terms of content. A comprehensive review of the relevant literature is lacking. Also, the result's section needs to be updated to provide additional discussion on the results they obtained.

I recommend **Moderate Revision**, and specifically addressing issues related to the introduction and the results section. | The authors wish to thank the reviewers for their time in effort in reviewing our manuscript.

We hope the changes listed have made the manuscript suitable for publication. |
| **Major Comments** | |
| **Introduction:**
1) The introduction is too short, and the body of the introduction section is only one paragraph, lacking sufficient background information.
2) Furthermore, a review of literature on the physical approaches for PMP estimation is lacking which can enhance the introduction. | 1) The introduction was revised. Background information was mentioned in the separate paragraph.

2) A review of literature on the physical approaches for PMP estimation was added to text. |
| The results section needs to be updated to add additional discussions and analyses on different aspects of the results they already obtained. Just providing one or two sentences that are obvious from the figures or tables is not appropriate for a paper at this level. | The results were written based on this comment. |
| An appendix section must be added to give a brief overview of the PMP Calculator app, along with its download link for the interested readers. | The appendix was available for this application, but according to the editor's opinion regarding the deletion of Figure 2, the appendix was not included in the article's text. Whenever a person wants software, send authors an email, and then we send him/her this application. |
| Authors are strongly encouraged to check the grammar and language of the manuscript before resubmission. Some of these errors are mentioned here, but there are more errors and typos that need to be corrected. | The manuscript was checked from the viewpoint if the grammar and language. |
| **Minor Comments** | |
| P1L19: Must be "rainfalls" | It was corrected and was mentioned in P1L24 by yellow highlight. |
| P1L19: delete "one" , replace with "among" | It was corrected and was mentioned in P1L24 by yellow highlight. |
| P1L19: What are social damages? Damages to societies seems to be more relevant here. | The sentence was revised to "Intensive rainfalls and heavy floods are among the most catastrophic natural hazards which have large social consequences for communities all over the world." |
| P1L22: Replace "specific project" with "hydrologic infrastructure" | It was corrected and was mentioned in P1L27 by yellow highlight. |
| P1L25: Where does the quotation end? | It was corrected and was determined in P1L30 by yellow highlight. |
| P1L26: "A statistical …" Review of statistical methods should go to a separate paragraph. | It was corrected and was determined in P2L3 by yellow highlight. |
| P1L29:30 years "of" daily data … | It was corrected and was determined in P2L4 by yellow highlight. |
| P1L28: "basins" | It was corrected and was determined in P2L5 by yellow highlight. |
| P1L29: "kilometers" | It was corrected and was determined in P2L5 by yellow highlight. |
| P1L29:
1) Mention the names of other statistical methods and discuss their differences.
2) Also mention why the Hershfield method is more popular.
3) Review of statistical methods must be a separate paragraph for itself. | 1) The name of the statistical methods was mentioned in P2L6, and was determined by yellow highlight.
2) It was mentioned in p2L18 and was determined by yellow highlight.
3) It was applied. |

| | |
|---|---|
| P2L1: Any references? | It was mentioned in P2L23.

Rakhecha, P. R., Deshpande, N. R., and Soman, M. K.: Probable maximum precipitation for a 2-day duration over the Indian Peninsula. Theor. Appl. Climatol., 45, 277-283, 1992. |
| P2L2: Discuss the physical methods in a separate paragraph. | It was discussed and was determined in P2L24 by yellow highlight. |
| P2L2: "characteristic of the deterministic…" Not sure what you mean! | It was deleted. |
| P2L6: Some physical methods are mentioned; however, nothing special about their characteristics and differences have been mentioned. The only thing mentioned is that they are not easy to use. Please consider adding more details about the different physical methods, and their differences and pros and cons. Also, keep in mind that the difficulty in estimation is not the case in many parts of the world. | Required descriptions were added to the text and were determined in P2L24 by yellow highlight. The advantages and disadvantaged of physical methods were discussed. |
| P2L8:

1) Comparison of the physical and statistical methods need to go to a separate paragraph.
2) A more detailed review of the literature is required. For instance, why in some regions the two methods give similar results and why in some other regions they are totally different? | 1) Comparison of the physical and statistical methods was mentioned in a separate paragraph and was determined in P3L7 by yellow highlight.

2) Some sentence was added about this difference (P3L20). |
| P2L11: "The results of these researchers have indicated that although the statistical approaches provide larger estimates of PMP, it is proposed for areas where hourly rainfall, dew point temperature, wind speed, and vertical radiosonde measurements are unavailable"

The first and the second statements are irrelevant to each other. | The sentence was revised and mentioned in P3L23. |
| P2L18-19: It has already been mentioned in L11. | The sentence in line 11 was mentioned with focus on the studies in Iran. |
| P2L19: What is the overall conclusion from these studies? | The overall conclusion was added in P3L34. |
| P2L20: Replace "was" with "is" | This part was rewritten and was replaced by new sentences. It was determined in P4L9 by yellow highlight. |
| P2L20: delete "written" and write "prepare a" instead | This part was rewritten and was replaced by new sentences. It was determined in P4L9 by yellow highlight. |
| P2L20: Estimation alone is not a good goal for a paper. You can draw more useful information from your results. For instance, using the PMP24 maps, you can specify the regions that are more likely to experience intense storms. Such information could be useful for water resources planning and management, flood risk assessment, and catastrophe management. | The aim of study was rewritten. It was determined in P4L9 by yellow highlight. |
| P3L2:

1) Where do you want to put figure 1?
2) For figure 1, it is also suggested to name each of the small figures, as a, b, and c. Then, give a short description in the figure caption for each of them. | 1) After the first paragraph of materials and methods in P4L26.
2) It was corrected based on the reviewer's comment. Required description was added in P16 by yellow highlight |
| P3L17: Delete "then" | It was deleted based on the reviewer's comment. |
| P3L18: To be added to what? | It was corrected and was determined in P5L10 by yellow highlight.

It means the number of standard deviations ($S_n$) to be added average rainfall ($\overline{X}_n$) to obtain PMP. It should be noted that, this sentence is from "Manual on Estimation of Probable Maximum Precipitation, page 66, section 4.2-1, Line 12".

If the maximum observed rainfall $X_m$ is substituted for $X_t$, and $K_m$ for $K$, $K_m$ is then the number of standard deviations to be added to to obtain $X_m$, or

$X_t = \overline{X}_n + K_m S_n$        (4.2) |
| P3L19: Delete "of America" | It was deleted. |
| P4L1: should be "the" standard deviation | It was corrected and was determined in P5L14 by yellow highlight. |
| P4L2: To do what? Say, the goal first; then, mention the steps. | It was corrected and was determined in P5L16 by yellow highlight. |
| P4L12: Information about the discharge data is still missing in the data section. Add it! | Maximum 24-hour rainfall in each station and the maximum instantaneous peak discharge were used to determine the date of storms. The maximum instantaneous peak discharge data was used because of the inherent dependence of rainfall and runoff. Therefore, The maximum instantaneous peak discharge has resulted from an intensive rainfall after a certain lag-time or delay. Based on the authors' opinion, because discharge data was not used directly in the calculation of PMP, was not mentioned in the text. |

| | |
|---|---|
| P5L12: mean "squared" error | It was corrected and was determined in P7L4 and P7L5 by yellow highlight. |
| P6L20:

1) Provide more discussion on figures. | P8L17:

1) A more detail analysis of the PMP in the study area could be presented using the $PMP_{24}$ isohyetal lines using the Hershfield standard and revised methods which are shown in Fig. 2. PMP values at each point in the study area could be approximated from these maps. Also, the range of PMP values and its variation was shown clearly. From Fig. 2(a), it is clear that the highest $PMP_{24}$ values for the standard Hershfield method is at the southwest parts of basin around the Kord Kooy and Ghaz Mahalleh stations which are from 450 to 430 mm, whereas the lowest PMP values is at the south-eastern parts of basin around the Ziarat where the isohyetal lines are less than 240 mm. From Fig. 2(b), the $PMP_{24}$ values resulting from the use of the Hershfield revised method are lower in south-eastern parts and higher in the western parts of the study area. |
| 2) Is there any specific gradient in the PMP values? | 2) Generally, the $PMP_{24}$ values resulting from both Hershfield methods decrease from west to east (Fig. 2). |
| 3) Which parts of the basin experience more severe storms? Which parts of the basin experiences more extreme precipitations? Why? | 3) The results of Fig. 2 showed that the western parts of the basin, that are closer to Caspian Sea, experience more severe storms. |
| 4) Is the basin homogenous in this regard? | 4) The studied stations are located in the homogeneous parts of Golestan province. This issue is confirmed in the article (N. Hasanalizadeh, N., Mosaedi, A., Zahiri, A.R., Babanezhad, M. 2014. Determine of Homogeneous Regions Dirstibution of Annual Rainfall in Golestan Province Using Clustering and L-moments. Journal of Water and Soil, 28(5): 1061-1071. [in Persian with English abstract]). |
| P8L14: Not sure what does the most moderate mean! | It means Lenient. It was corrected and was determined in P10L19 by yellow highlight. |
| P9L4: "Based on …." not relevant to the previous sentences. It could be a separate paragraph, joint with the next paragraph. | It was corrected and was determined in P11L5 by yellow highlight. |

**Appendix**

**Introduction**

PMP calculator is a user friendly and multi-platform application tool dedicated to the estimation of Probable Maximum Precipitation (PMP) for 5 minutes, 1, 6 and 24-hours duration using the Hershfield standard method (proposed by WMO, 2009) and the Hershfield revised method by Dasa et al. (2001). In this application, PMP is calculated without consideration of adjustment of area reduction curve and depth area relation.

**Installation**

PMP Calculator is a user-friendly and multi-platform JAVA application which is applicable on Windows, Linux, and Macintosh OS X platforms.
Run "pmp.jar" application to initialize the software setup.

**Data Input**

The Input data for both methods are the annual maximum precipitation for a certain duration including 5 minutes, 1, 6 and 24 hour. In order to improve the interface of the application the input files are in MS Excel worksheet format (Fig. 1).

[Figure]

**Fig. 1. Example of the file format**

**Calculation Process**

The setting for the calculation of PMP values are defined in the PMP calculator by pressing "Select File…" (Fig. 2). Then, user should select the duration (Fig. 3). The output file is stored at the input folder. The output file is contained three sheets, which input data is stored on the first sheet, the results of the standard method is stored in second sheet, and the results of the revised method is stored in third sheet.

[Figure]

**Fig. 2. Select the input file**                    **Fig. 3. Select the duration**

**Output data**

In the Hershfield standard method, output file has included in N, mean, $K_m$, SD, Max, $Mean_{n-m}$, $SD_{n-m}$, E, F, $C_1$, $C_2$, $C_3$, $C_4$, $Mean_1$, $SD_1$, $PMP_{ini}$, $C_5$, $PMP_{final}$ and CP. Description of output data in the Hershfield standard method is mentioned in Table 1. The output file is shown in Fig. 4.

**Table 1. Description of output data in first approach**

| variable | Description |
|---|---|
| N | Length of record |
| Mean | Mean of the annual series |
| $K_m$ | Frequency factor |
| SD | Standard deviation of the annual series |
| Max | The maximum item in the series |
| $Mean_{n-m}$ | Mean of the annual series computed after excluding the maximum item in the series |
| $SD_{n-m}$ | Standard deviation of the annual series computed after excluding the maximum item in the series |
| E | Ratio of the $mean_{n-m}$ to the mean |
| F | Ratio of the $SD_{n-m}$ to the SD |
| $C_1$ | Adjustment of $\bar{X}_n$ for maximum observed event |
| $C_2$ | Adjustment of $S_n$ for maximum observed event |
| $C_3$ | Adjustment of $\bar{X}_n$ for sample size |
| $C_4$ | Adjustment of $S_n$ for sample size |
| $Mean_1$ | Adjusted $\bar{X}_n$ |
| $SD_1$ | Adjusted $S_n$ |
| $PMP_{ini}$ | Initial PMP |
| $C_5$ | Adjustment for fixed observational time intervals |
| $PMP_{final}$ | Final PMP |
| CP | Ratio of the PMP to the maximum item in the series |

In the Hershfield revised method, output file has included in N, Mean, SD, Max, $Mean_{n-m}$, $SD_{n-m}$, $K_m$, $K_m^*$, $PMP_{ini}$, C, $PMP_{final}$ and CP. Description of output data in the Hershfield revised method is mentioned in Table 2. The output file is shown in Fig. 5.

**Table 2. Description of output data in second approach**

| variable | Description |
|---|---|
| N | Length of record |
| Mean | Mean of the annual series |
| SD | Standard deviation of the annual series |
| Max | The maximum item in the series |
| $Mean_{n-m}$ | Mean of the annual series computed after excluding the maximum item in the series |
| $SD_{n-m}$ | Standard deviation of the annual series computed after excluding the maximum item in the series |
| $K_m$ | Frequency factor |
| $K_m^*$ | The maximum Frequency factor |
| $PMP_{ini}$ | Initial PMP |
| C | Adjustment for fixed observational time intervals=1.13 |
| $PMP_{final}$ | Final PMP |
| CP | Ratio of the PMP to the maximum item in the series |

[Figure]

Fig. 4. The output of Hershfield standard method

[Figure]

Fig. 5. The output of Hershfield revised method

---

## Author Response (AR4)

**Probable Maximum Precipitation Estimation in a Humid Climate**

Dear Editor,

The authors appreciate the editors and reviewers. The manuscript carefully was checked from the grammatical point of view. Also, all changes were determined in the main text by yellow highlight.

With warm regards, Corresponding Author, Bakhtiari, B.